# Graph Theory-Based Deep Graph Similarity Learning: A Unified Survey of Pipeline, Techniques, and Challenges

**Zhouyang Liu**                                                                                   *liuzhouyang20@nudt.edu.cn*
*College of Computer Science and Technology*
*National University of Defense Technology*

**Ning Liu**[†]                                                                                    *liuning17a@nudt.edu.cn*
*College of Information and Communication*
*National University of Defense Technology*

**Yixin Chen**                                                                                     *chenyixin@nudt.edu.cn*
*National Key Laboratory of Parallel and Distributed Computing*
*National University of Defense Technology*

**Ziqing Wen**                                                                                     *zqwen@nudt.edu.cn*
*College of Computer Science and Technology*
*National University of Defense Technology*

**Jiezhong He**                                                                                    *jiezhonghe@nudt.edu.cn*
*College of Computer Science and Technology*
*National University of Defense Technology*

**Dongsheng Li**[†]                                                                                *dsli@nudt.edu.cn*
*National Key Laboratory of Parallel and Distributed Computing*
*National University of Defense Technology*

**Reviewed on OpenReview:** *https://openreview.net/forum?id=fHf4jbIfex*

## Abstract

Graph similarity computation, which measures the resemblance between graphs, is a crucial operation in fields such as graph search. Recent advances in graph neural networks have enabled the embedding of graphs into low-dimensional vector spaces, where the similarity or distance between graphs can be efficiently quantified. However, these methods are often tailored to specific tasks and function as black boxes, limiting both generalization and interpretability. To address these challenges, there is growing interest in incorporating domain-agnostic and interpretable concepts from graph theory—such as subgraph isomorphism, maximum common subgraph, and graph edit distance—into graph similarity learning as training objectives. This survey presents a comprehensive review of recent advancements in deep graph similarity learning, focusing on models that integrate these graph theory concepts. Despite the different training objectives of these approaches, they share significant commonalities in the training pipeline, techniques, and challenges. We analyze them within a unified lens referred to as graph theory-based deep similarity learning (GTDGSL) methods. We systematically compare existing GTDGSL methods alongside their common training pipeline, highlighting the technique trend and discussing key challenges, applications, and future research directions in this domain. We organize the papers included in this survey and their open-source implementations at https://github.com/liuzhouyang/Graph-Theory-Based-Deep-Graph-Similarity-Learning-Survey.

# 1 Introduction

Entities can be represented as nodes within a graph, with edges depicting their relationships. This graph-based representation effectively captures the complex interactions present in real-world systems, ranging from social networks to biochemical molecule structures. This representation, in turn, facilitates the identification of similar interaction patterns, i.e., structural similarities between graphs. These structural similarities often correlate with functional similarities, highlighting the importance of graph similarity learning in exploring these relationships.

Such similarity measurement serves as a fundamental operation in various downstream tasks, including graph classification (Mohamed et al., 2019), clustering (Liu et al., 2024), and search (Zheng et al., 2014; He & Singh, 2006; Wang et al., 2010; Zhu et al., 2012; Zheng et al., 2013). For instance, in bioinformatics, identifying similar structures can lead to the discovery of new drugs or treatments (Schadt et al., 2009). In social network analysis, graph similarity is crucial for detecting communities, identifying influencers, and understanding the spread of information (Narayan & Kumar, 2016). Similarly, in recommendation systems and fraud detection, graph similarity plays a key role in matching user preferences and identifying anomalous patterns (Kim et al., 2022).

As graph similarity learning has been studied for several decades, many techniques have been proposed, such as kernel-based methods (Borgwardt & Kriegel, 2005; Costa & Grave, 2010; Shervashidze et al., 2011; Kriege, 2022; Nikolentzos et al., 2022) and graph spectral-based methods (ElGhawalby & Hancock, 2008; Crawford et al., 2017). However, these methods often rely on predefined patterns or structures to represent graphs. Recent advances in graph neural networks (GNNs) have empowered models to learn abstract representations that capture the most relevant substructures across diverse graphs according to the downstream task. This shift reduces dependence on hand-crafted features, enhances the generalizability of the methods, and broadens the design space for model architectures and training paradigms.

Based on deep learning techniques such as graph neural networks (GNNs) (Kipf & Welling, 2017; Xu et al., 2019; Veličković et al., 2018; Brody et al., 2022), deep graph similarity learning (DGSL) methods project graphs into low-dimensional vector spaces, where the distances between graph pairs effectively capture and reflect their structural similarities and differences. More detailed surveys on general DGSL can be found in the work of Ma et al. (2019); Ju et al. (2024). However, the distance measurements produced by these models are often task-specific and can be difficult to interpret, complicating the understanding of the models' behaviors. In contrast, classical graph theory provides well-defined and domain-agnostic concepts of graph structural similarity through various problems. This survey focuses on three of them that are widely applied in graph search, which range from the stringent subgraph isomorphism (SI)—which is NP-complete (Garey & Johnson, 1979) and involves determining whether one graph is equivalent to a subgraph of another—to more flexible approaches that are robust to noise in real-world scenarios, such as the maximum common subgraph (MCS) and graph edit distance (GED). While both MCS and GED are NP-hard problems (Hjorth, 2005), they provide useful frameworks for evaluating similarity: MCS identifies the largest subgraph common to both graphs, while GED quantifies the minimum cost required to transform one graph into another through a sequence of edits.

Although these three problems are theoretically significant and form a basis for understanding and quantifying graph similarities, their inherent computational complexity makes them notoriously difficult to solve. Computing these concepts typically requires establishing a bijective mapping between elements, such as nodes and edges, across graph pairs that satisfy certain criteria. For example, in graph edit distance (GED) computation, the total edit cost to transform the source graph into the target graph must be optimized, necessitating exhaustive combinatorial enumeration. Existing efforts have proposed strategies for ordering the search process and pruning unpromising search branches to reduce the search space (Abu-Aisheh et al., 2015; Chang et al., 2020; McCreesh et al., 2017; Carletti et al., 2018). However, these methods operate in an on-the-fly manner, which means each computation relies on a specific graph pair, rendering intermediate results from one search inapplicable to another, impacting the efficiency of these methods.

Recently, several end-to-end learning-based methods (Bai et al., 2019; Ying et al., 2020; Doan et al., 2021) have proposed leveraging distances in embedding spaces to reflect the similarity between graph pairs based on

the aforementioned graph theory concepts. In addition, some methods operate in learn-to-search scenarios, enhancing the searching process of conventional algorithms with pretrained deep learning models (Yang & Zou, 2021; Wang et al., 2021; Bai et al., 2021; Wang et al., 2022; He et al., 2022). These approaches not only offer clearer insights into the structural relationships between graphs but also significantly accelerate the computation of these complex graph theory problems. In recent years, various works have followed this trend and further refined existing methods, leading to notable advancements in both accuracy and efficiency (Zhuo & Tan, 2022; Liu et al., 2023b; Roy et al., 2022a). Although these methods are tailored to different problems, this paper analyzes them through a unified lens, systematically presenting their theoretical connections, technical similarities, and differences. Moreover, this unified view enables a more comprehensive discussion of common challenges, such as preserving graph characteristics and ground-truth acquisition problems, etc. Finally, this perspective facilitates the identification of future research directions and broadening the field.

**Scope and contributions.** Unlike previous surveys on general deep graph similarity learning or deep similarity learning, such as Ma et al. (2019); Ju et al. (2024); Yang et al. (2024), which primarily focus on model taxonomies, general GNN architectures, learning paradigms or data-specific discussions, this survey distinguishes itself in two key aspects. First, it specifically focuses on deep graph similarity learning approaches that approximate three selected graph theory concepts for quantifying graph similarity: subgraph isomorphism, maximum common subgraph, and graph edit distance, collectively referred to as Graph Theory-based Deep Graph Similarity Learning (GTDGSL). By examining these approaches under a unified lens, this survey highlights their commonalities and differences in both theoretical foundations and technical implementations. Second, it provides an in-depth review of GTDGSL-related techniques, emphasizing technical developments and trends that have been underexplored in prior works. To the best of our knowledge, this is the first survey dedicated specifically to the GTDGSL problem. The main contribution of this survey can be summarized as follows.

- We briefly summarize three graph theory concepts—subgraph isomorphism, maximum common subgraph, and graph edit distance—and analyze their theoretical connections, along with the technical similarities and differences among conventional algorithms for these problems.

- We categorize existing GTDGSL methods alongside their training pipeline, establishing a unified framework for systematic comparison.

- We analyze and review GTDGSL methods at each step of the training pipeline to align their inputs and outputs, highlight technical trends at each step, and elucidate their design space to provide technical insights.

- We provide a detailed discussion of dataset generation, evaluation metrics, and their downstream applications.

- We identify key challenges and opportunities for future research in this domain.

**Organization.** This paper is organized as follows. In Section 2, we provide the necessary background on the GTDGSL problem, covering key graph theory concepts, traditional techniques, as well as a formal problem formulation of GTDGSL. We also compare GTDGSL with conventional algorithms and DGSL, offering a critical analysis of its advantages, disadvantages, and distinct model designs. In Section 3, we categorize and review the current GTDGSL methods approximating SI, MCS, and GED, alongside their common training pipeline, revealing the design space and technique trends of GTDGSL methods. In Section 4, we introduce the dataset generation techniques in existing work. In Section 5, we present the metrics evaluating GTDGSL models. In Section 6, we discuss the applications of GTDGSL methods. Finally, in Section 7, we identify the key challenges shared among GTDGSL methods and highlight the future directions. The survey is briefly concluded in Section 8.

## 2 Background

In this section, we begin with the notations that will be used throughout this survey. Next, we provide formal definitions for the three selected graph theory concepts—subgraph isomorphism, maximum common

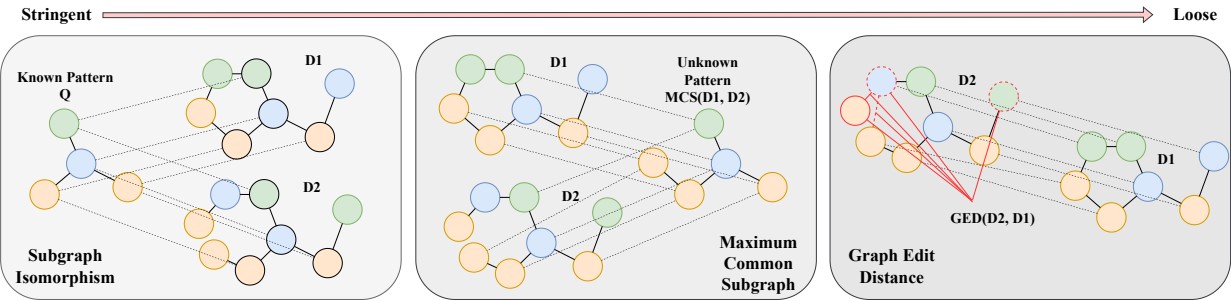

Figure 1: A toy example illustrating the three selected graph theory concepts. Black dashed lines represent mappings, while red markings indicate edits. Subgraph isomorphism verifies the presence of a known pattern within a larger graph. MCS identifies the unknown largest connected common subgraph between two graphs. GED transforms one graph into another with the minimum edit cost.

subgraph, and graph edit distance—along with their distinct properties and a brief overview of conventional solutions. Following this, we explore the connection between graph neural networks (GNNs) and the computation of these concepts, laying the foundation for graph theory-based deep graph similarity learning (GTDGSL). We also compare GTDGSL with conventional algorithms and general graph similarity learning, offering a critical analysis of its advantages, disadvantages, and distinct model designs.

**Notation.**   This survey focuses on node-labeled undirected graphs. For a given graph $G = (\mathcal{V}_G, \mathcal{E}_G)$, where $\mathcal{V}_G$ represents the set of nodes and $\mathcal{E}_G$ denotes the collection of edges $e(v, u)$ for $v, u \in \mathcal{V}_G$, each node $v \in \mathcal{V}_G$ is associated with a feature vector $\mathbf{x}^0 \in \mathbf{X}$, with $\mathbf{X}$ being the collection of node features. This notation can be easily extended to unlabeled graphs by assigning identical features to each node. Since $G$ is undirected, for every edge $e(v, u) \in \mathcal{E}_G$, there exists a counterpart $e(u, v)$. The cardinality of the node and edge sets are denoted by $|\mathcal{V}_G|$ and $|\mathcal{E}_G|$, respectively. The sparse edge set $\mathcal{E}_G$ can be represented as a dense adjacency matrix $\mathbf{A}^{|\mathcal{V}_G| \times |\mathcal{V}_G|}$, where $\mathbf{A}(v, u) = 1$ if $e(v, u) \in \mathcal{E}_G$ and $\mathbf{A}(v, u) = 0$ otherwise, with $\mathbf{A}(v, u)$ representing the entry at row $v$ and column $u$. A connected subgraph $G' = (\mathcal{V}_{G'}, \mathcal{E}_{G'})$ of $G$ consists of subsets of $\mathcal{V}_G$ and $\mathcal{E}_G$ such that for each edge $e(v, u) \in \mathcal{E}_{G'}$, both $v$ and $u$ belong to $\mathcal{V}_{G'}$, and all vertices in $G'$ are reachable from one another by traversing edges within $\mathcal{E}_{G'}$.

## 2.1   Graph Similarity Concepts in Graph Theory

In the following, we will present formal definitions of selected graph theory concepts, and discuss their distinct properties as well as their conventional solutions, emphasizing both their similarities and differences to deepen the understanding of their interconnectedness.

### 2.1.1   Subgraph Isomorphism (SI)

**Definition.**   Given a query graph $Q$ and a data graph $D$, if $Q$ is isomorphic to a subgraph $D'$ of $D$, there exists at least one bijective function $f$ between $Q$ and $D'$, such that $f : \mathcal{V}_Q \mapsto \mathcal{V}_{D'}$ satisfying (1) for all $v \in \mathcal{V}_Q$ and $v' \in \mathcal{V}_{D'}$, there exists $f(v) = v'$ and $\mathbf{x}_v^0 = \mathbf{x}_{v'}^0$. (2) For all $e(v, u) \in \mathcal{E}_Q$, there exists $e(f(v), f(u)) \in \mathcal{E}_{D'}$. We denote $(Q, D)$ as a subgraph isomorphism. In this context, graph isomorphism is a special case of subgraph isomorphism where $D' = D$. Another related variant of subgraph isomorphism is subgraph edit distance (SED), which depicts the minimum edit cost for transforming a graph into a subgraph of another. The SED for matched pair $(Q, D)$ equals zero; otherwise, it would be a non-negative value.

**Properties.**   SI defines a partial ordering relationship between graphs, characterized by the following properties (Ying et al., 2020).

- *Transitivity.* If $Q$ is a subgraph of $D'$ and $D'$ is a subgraph of $D$, then $Q$ is a subgraph of $D$.

- *Anti-symmetry.* If $Q$ is subgraph of $D$, $D$ is a subgraph of $Q$, then $Q$ and $D$ are isomorphic.

- *Intersection set.* The set of common subgraphs of $D_1$ and $D_2$ is contained within the intersection of their respective subgraph sets.

The first two properties make SI a partial order, while the last property helps illustrate the connection between SI and the Maximum Common Subgraph, which we will discuss later.

**Conventional Solutions.** According to downstream scenarios, conventional solutions for subgraph isomorphism can be divided into two groups: subgraph enumeration (also known as subgraph matching) and subgraph indexing. Subgraph enumeration methods aim to exhaustively enumerate all bijective mappings (isomorphisms) from the query graph to subgraphs within the data graphs (He & Singh, 2008; Shang et al., 2008; Carletti et al., 2018; Archibald et al., 2019; Han et al., 2013; 2019; Sun et al., 2020). State-of-the-art methods in this group typically follow a preprocessing-ordering-enumeration paradigm, they first apply heuristic rules to filter out unpromising vertices and edges based on the graph pairs, then define an optimal search order to improve the efficiency of the enumeration process. For a detailed review of these methods, we refer readers to Sun & Luo (2020); Zhang et al. (2024b).

In contrast to these on-the-fly enumeration methods, subgraph indexing methods rely on predefined subgraph patterns (i.e., features) to construct an index for the data graphs (Zhang et al., 2009; Zhao & Han, 2010; Klein et al., 2011; Xie & Yu, 2011; Giugno et al., 2013; He et al., 2024). These methods typically count the occurrences of specific subgraph patterns within each graph and use these counts to build an index. Intuitively, $(Q, D)$, the index of $D$ on any feature should contain higher or equal counts compared with $Q$'s, which can be refer to as subgraph containment constraint. Given a query graph and a collection of data graphs, indexing methods aim to filter out unpromising data graphs by comparing the indices of graph pairs.

### 2.1.2 Maximum Common Subgraph (MCS)

**Definition.** The graph $M$ is the maximum common connected subgraph between $Q$ and $D$, if $M$ satisfying (1) $(M, Q)$ and $(M, D)$ are subgraph isomorphisms, (2) $M$ has the most nodes compared with other common subgraphs exist $Q$ and $D$. This definition highlights the connection between SI and MCS, as MCS represents the largest common subgraph, measured by the number of nodes or edges, within the *intersection set* of $Q$ and $D$.

**Properties.** MCS has the following properties.

- *Non-transitivity.* If $M_1 = \text{MCS}(Q_1, D)$ and $M_2 = \text{MCS}(Q_2, D)$, it does not necessarily mean that $M_1, M_2$ or that the MCS of $Q_1$ and $Q_2$ includes either $M_1$ or $M_2$.

- *Symmetry.* For any two graphs $Q$ and $D$, $\text{MCS}(Q, D) = \text{MCS}(D, Q)$.

**Conventional Solutions.** Conventional Maximum Common Subgraph (MCS) algorithms typically employ backtracking, constraint programming, or clique-based reduction as search strategies (McGregor, 1981; Raymond & Willett, 2002; Ehrlich & Rarey, 2010; Ndiaye & Solnon, 2011; McCreesh et al., 2016; 2017). State-of-the-art algorithms address the MCS problem using a branch and bound optimization approach (McCreesh et al., 2017). This methodology systematically explores the search space by recursively partitioning it into smaller subproblems, then estimates an upper bound on the maximum possible size that can be formed from the current partial mappings to prune unpromising branches.

### 2.1.3 Graph Edit Distance (GED)

**Definition.** The graph edit distance $\text{GED}(Q, D)$ is the minimum cost of the edit operation sequence that transforms $Q$ into $D$. The viable edit operations are node/edge insertion, deletion, and label substitution. Each operation can be assigned with a distinct non-negative cost, and different cost settings can produce different GEDs.

**Properties.** Under a uniform cost setting, where each operation has the same cost, GED is a metric and satisfies the following properties (Ranjan et al., 2022).

- *Symmetry.* For any two graph $Q$ and $D$, $\text{GED}(Q, D) = \text{GED}(D, Q)$, though the specific edit sequences may differ.

- *Triangle Inequality.* $\text{GED}(Q, D) \leq \text{GED}(Q, D_1) + \text{GED}(D_1, D_2)$.

However, when under a non-uniform cost settings, the above properties are no longer guaranteed, and GED behaves as a non-metric distance function with the following properties.

- *Asymmetry.* $\text{GED}(Q, D) \neq \text{GED}(D, Q)$.

Specifically, when node deletion and insertion each have a cost of one, label substitutions do not occur, and edge insertion or deletion is free, the GED computation becomes equivalent to the MCS problem (Bunke, 1997).

**Conventional Solutions.** Conventionally, the exact GED problem is often formulated as a path-finding problem (e.g., A* search). In contrast, the approximation of GED is regarded as a quadratic assignment problem or bipartite graph matching problem (Bunke, 1997; Zeng et al., 2009; Riesen & Bunke, 2009; Riesen et al., 2013; Abu-Aisheh et al., 2015; Bougleux et al., 2017; Stauffer et al., 2017; Blumenthal & Gamper, 2020; Chang et al., 2020). These methods often require padding the smaller graph between $Q$ and $D$ with dummy node to ensure they have the same size. They then create a cost matrix that quantifies the differences between each cross-graph node (or edge) pair. Based on the cost matrix, the exact ones gradually expand the search path till each node in $Q$ is matched to a node in $D$ or a dummy node, while the approximate ones directly construct node mapping from $Q$ to $D$ to minimize the total cost.

| | SI | MCS | GED |
|---|---|---|---|
| Prior Knowledge | Yes | No | No |
| Largest | No | Yes | No |
| Connected | Yes | Yes | No |
| Preprocessing | Graph property-based | Graph property-based | Score-based |
| Ordering | Static; Graph property-based | Static; Graph property-based | Dynamic; Score-based |
| Pruning/Bounding | Hard constraints | Score and hard constraints | Score |

Table 1: Connections Between the Selected Graph Theory Concepts

### 2.1.4 Connections Between Graph Similarity Concepts

**Theoretical Connections.** All three concepts are NP-hard problems and can be understood as attempts to establish (partial) node mappings between graph pairs. Each of them allows multiple possible optimal mappings, but all lead to the same final result. Moving from SI to MCS to GED, the constraints on structural properties become progressively more relaxed. This relaxation trend makes GED more general but also more challenging to optimize. SI verify the presence of $Q$ in $D$, meaning the common structure is predefined as $Q$. In contrast, MCS does not assume prior knowledge of a shared structure but instead seeks to discover the largest common subgraph between the given graph pair within the intersection set. GED, also considering common substructures, but it neither requires them to be the largest nor necessarily connected, as it determines preserved substructures based on the cost of edit operations rather than explicit structural constraints. Additionally, under special cost setting, computing GED is equivalent to MCS problem (Bunke, 1997). See Table 1 for summarization.

**Technical Similarities and Differences.** These concepts share key search processes, such as preprocessing, ordering, and pruning/bounding, but they adopt distinct strategies for each step. We summarize these differences in Table 1 and provide a detailed discussion below.

- *Preprocessing.* SI and MCS algorithms typically perform a preprocessing step to reduce the candidate space of query and data nodes by eliminating infeasible matches. Common techniques include degree filtering and label filtering, which leverage graph properties to constrain the search space. In contrast, GED algorithms estimate lower bounds using linear assignment solvers, such as the Hungarian algorithm.

- *Ordering.* SI and MCS require an explicit ordering before search to guide partial mapping expansion, typically relying on graph properties such as vertex degrees, label frequency, or connectivity. In contrast, GED algorithms is cost-sensitive, dynamically adjust the search order based on estimated edit costs.

- *Pruning/Bounding.* SI enforces strict constraints, such as degree filtering, connectivity checks, and label consistency, to efficiently prune infeasible mappings. In contrast, GED and MCS algorithms rely more on bounding functions rather than hard constraints to eliminate partial matches that cannot improve the best-known score. Nonetheless, MCS may still enforce structural feasibility constraints, such as connectivity.

## 2.2 Graph Theory-Based Deep Graph Similarity Learning (GTDGSL)

Deep graph similarity learning, empowered by deep learning techniques, aims to construct a similarity function that assigns meaningful similarity scores to graph pairs. Graph Neural Networks (GNNs) naturally emerge as a key tool in this context, as they effectively generate node, edge, and graph representations while considering the connectivity patterns of graphs.

### 2.2.1 Graph Neural Networks

Message-passing Graph Neural Networks (MPNNs), commonly referred to as GNNs, focus on capturing the local structure around each node, where the computational graph of a node can be viewed as a node-anchored $k$-order Weisfeiler-Lehman (WL) subtree, which grows according to the edge-defined $k$-hop accessible neighborhood of each node. Specifically, they generate node embeddings for each graph using an aggregation-update paradigm. For a node $v \in \mathcal{V}$ with an initial feature vector $\mathbf{x}_v^0$, the process involves aggregating features from its neighboring nodes according to the adjacency matrix of the graph, and then updating the feature of $v$, this process at $l$-th layer can be described as follows:

$$\mathbf{x}_v^l = \text{Update}(\mathbf{x}_v^{l-1}, \text{Aggr}(\mathbf{x}_u^{l-1} : u \in \mathcal{N}_v)),$$

where $\mathbf{x}_v^l$ is the embedding of node $v$ at the $l$-th layer, and $\mathcal{N}_v$ denotes the set of neighboring nodes of $v$. The function $\text{Aggr}(\cdot)$ aggregates the features of the neighboring nodes $u$, and the $\text{Update}(\cdot, \cdot)$ function updates the feature of the node $v$ based on its previous feature and the aggregated information. Since graphs are non-Euclidean data structures without a natural order, and the number of neighbors for each node can vary, GNNs typically aggregate and update node features in a permutation-invariant manner, ensuring the robustness to node reordering. Xu et al. (2019) further propose projecting the updated features using a Multilayer Perceptron (MLP), which consists of multiple fully connected layers, with a nonlinear activation function applied between each pair of linear layers. This projection ideally ensures the injectiveness of the embeddings, thereby enhancing their expressiveness. The outputs of a $k$-layer GNNs for each node can be further summarized as the graph representation with a pooling function, such as a dimensional-wise sum pooling. Despite their advantages in graph handling, GNNs face several challenges and limitations which we briefly discuss as follows.

- *Over-smooth.* As the aggregation-update process is repeated over multiple layers, the $k$-hop neighborhoods of different nodes begin to overlap more extensively. As a result, node features become

increasingly similar, eventually converging to indistinguishable embeddings (Li et al., 2018). This issue is particularly pronounced in deep GNN architectures, where the loss of distinct node characteristics can severely degrade performance. While techniques such as skip connections (Chen et al., 2020) have been proposed to mitigate this issue, they often require careful tuning and may not fully resolve the problem in highly dense graphs.

- *1-WL Test Bounded Expressiveness.* The expressiveness of widely used GNNs is limited by the 1-Weisfeiler-Lehman (1-WL) test (Xu et al., 2019). This limitation manifests when nodes with distinct neighborhood structures (e.g., a triangle and a 4-node cycle) are assigned identical representations due to their isomorphic WL subtrees. Such cases are common in real-world graphs. Although augmenting node features with unique identifiers (You et al., 2021) or random features (Sato et al., 2021) can improve expressiveness, these approaches often initialization sensitive, which may degrade the robustness and generalizability of GNNs. More complex GNN variants, such as Subgraph GNNs (Zhang & Li, 2021) and High-order GNNs (Morris et al., 2019; Zhang et al., 2024a), can offer enhanced expressiveness but at the cost of significantly higher computational complexity, making them less practical for large-scale applications.

- *Loss of Overall Structure.* GNNs operate locally, aggregating information from immediate neighbors, which can lead to the loss of global structural information. This arises issues such as the automorphic node problem (Chamberlain et al., 2023), where nodes with isomorphic neighborhoods have identical representations. For tasks such as GTDGSL, which require alignment of global graph structures, this issue can severely hinder performance. While incorporating structural features, such as random walk embeddings (Grover & Leskovec, 2016), can partially address this limitation, it remains an open challenge to design GNNs that effectively balance local and global structural information.

- *Lack of Scalability.* The scalability of GNNs is another critical limitation, particularly when applied to large graphs with millions of nodes and edges. The aggregation-update process requires storing adjacency matrices and intermediate node/edge features, leading to high memory and computational demands. While methods such as graph sampling (Zeng et al., 2019) and graph partitioning can improve scalability, they often trade off accuracy for efficiency. For instance, sampling-based approaches may miss important structural information, while partitioning methods can introduce artifacts at graph boundaries. Decoupling feature transformation from propagation (Wu et al., 2019) has shown promise in reducing computational costs, but its effectiveness varies across different graph types and tasks.

For a more comprehensive discussion of these challenges and solutions, we refer readers to Wu et al. (2021); Chamberlain et al. (2023); Zhang et al. (2024a); Shao et al. (2024). In the following sections, we also highlight the solutions employed by existing GTDGSL methods in handling the limitation of GNNs, organized according to the training pipeline in Section 3. Furthermore, we analyze the specific challenges relative to GTDGSL and examine how existing approaches address these challenges in Section 7.

### 2.2.2 Problem Formulation

**General Framework.** Given any graph pair $(G_i, G_j)$, $D(\cdot, \cdot)$ is a learnable pairwise similarity function, such that $D(\phi(G_i), \phi(G_j)) \mapsto d_{ij} \in \mathbb{R}$, where $\phi$ is a projection function, such as GNNs, that transforms graphs into low-dimensional embeddings. Unlike general deep graph similarity learning, GTDGSL requires $d_{ij}$ to approximate specific graph theory targets, such as those defined above.

**Subgraph Isomorphism Prediction.** In the GTDGSL framework, subgraph isomorphism prediction at the graph level can be framed as a binary classification task, where the model outputs 1 if $(Q, D)$ is a subgraph isomorphism and 0 otherwise. Certain methods also propose to predict such subgraph isomorphism at the node level, they extract the $k$-hop subgraph $\mathcal{N}^k$ induced by a given central node, predicting whether $(\mathcal{N}^k(v), \mathcal{N}^k(u))$ is a subgraph isomorphism, where $v \in Q$ and $u \in D$.

**MCS and GED Approximations.** The predictions of MCS and GED are often formulated as a regression task. For MCS problem, the training objective of a GTDGSL model is the number of nodes within $M$ (ranging

from 0 to infinity) or the normalized value $\frac{|\mathcal{V}_M|}{(|\mathcal{V}_Q|+|\mathcal{V}_D|)/2}$ (ranging from $[0,1]$). Similarly, for GED problem, the target value is the non-negative GED value or the normalized GED value $e^{-\frac{GED(Q,D)}{(|\mathcal{V}_Q|+|\mathcal{V}_D|)/2}}$ to ensure a range of $(0,1]$.

### 2.3 GTDGSL vs. Conventional Algorithms

We compare conventional algorithms and GTDGSL across several key aspects:

**Correctness.** Conventional methods provide exact solutions, ensuring correctness by systematically searching the solution space. In contrast, GTDGSL methods approximate solutions and may suffer from false negatives, missing valid matches due to their learned representations (He et al., 2024). This trade-off makes GTDGSL more suitable for applications where approximate solutions suffice but limits their use in scenarios demanding provable guarantees.

**Efficiency and Scalability.** The efficiency and scalability of conventional algorithms vary significantly between datasets and tasks. SI algorithms have been demonstrated to scale to graphs with billions of edges. In contrast, exact MCS and GED algorithms suffer from exponential search complexity, causing memory overflow and impractical runtime for small graphs. GTDGSL models, on the other hand, have a different scalability limitation. Rather than search complexity, their scalability is primarily constrained by the GNN backbone. Large-scale graphs can lead to high computational and memory costs during message passing, making efficient GNN architectures crucial for scalability.

**Generalization Ability.** Conventional algorithms rely on heuristic optimization. They leverage predefined shallow information-based measures such as degree similarity, label frequency, and linear assignment solver-based lower bound estimation to guide the preprocessing, pruning, or bounding process. While effective heuristics such as maximum clique detection, can incur higher computational overhead, balancing effectiveness and efficiency. Furthermore, the performance of these heuristics is highly dependent on dataset characteristics, which typically do not generalize well across different datasets without manual tuning. In contrast, GTDGSL methods provide adaptive solutions without requiring extensive manual heuristics. They learn optimization strategies from data and can generalize across datasets, potentially discovering more efficient strategies than manually designed heuristics. However, this comes at the risk of overfitting to the training distribution, hurting robustness on unseen graphs.

**Intermediate Result Reusability.** Conventional algorithms generally compute results on demand, meaning that intermediate results from one query graph cannot be reused for another. This results in redundant computations when handling similar queries. GTDGSL models, however, can reuse learned representations for computed queries, reducing redundant computation.

**Training and Inference Time.** Conventional algorithms require no training but often incur high inference costs, especially on large graphs. GTDGSL models, in contrast, require substantial upfront training, but offer fast inference once trained. This makes GTDGSL advantageous for scenarios that require large amounts of similarity computations, where amortizing training costs over many queries is feasible.

### 2.4 GTDGSL vs. Deep Graph Similarity Learning (DGSL)

GTDGSL focuses on pairwise graph similarity, primarily based on graph structure, node/edge features, and graph theory constraints. This makes GTDGSL methods sensitive to small differences between graphs, ensuring that the results align with the properties of the targeted concept. In contrast, DGSL methods learn graph similarity in a task-driven manner, focusing on category-level similarity. The goal is to determine whether graphs belong to the same category by identifying distinguishing features across different categories. These distinct goals lead to different model designs. Next, we discuss their differences from two perspectives: model architecture and training sample generation methods. We summarize these differences and effective design choices in Table 2.

**Model Architecture.**   We analyze the differences between GTDGSL and DGSL in model architecture from three aspects:

- *Model Input.* GTDGSL focuses on computing the relative similarity between graph pairs, which can be viewed as finding a optimal (partial) mapping between graphs. In contrast, DGSL focuses on the properties of individual graphs. This distinction leads to differences in model input formats. GTDGSL typically takes two graphs as input, whereas DGSL processes graphs individually. To enhance effectiveness, some GTDGSL methods augment the initial node or graph features with pair-dependent heuristics to provide task-specific node compatibility information.

- *Graph Interactions.* GTDGSL often requires cross-graph interactions to capture similarity between graph pairs. It employs node/graph comparisons or cross-graph node/graph fusions to incorporate contextual information from another graph, aiding in the alignment of structural and feature information. The choice of interaction method involves trade-offs in terms of efficiency, effectiveness, and indexability. In contrast, DGSL processes a single graph without direct interaction with others.

- *Model Output.* The output of GTDGSL is a pair-dependent similarity score. In contrast, DGSL produces a likelihood for each possible label for a given graph.

**Training Sample Generation Method.**   We discuss the differences between GTDGSL and DGSL in this perspective from the following aspects:

- *Training Data.* The training data for GTDGSL can be sourced from graphs or sampled graphs across various domains, as GTDGSL is intended to be domain-agnostic. Graph samples for GTDGSL are typically generated through methods like BFS, DFS, or random walk-based traversals. For training graph pairs, similarity is often computed using conventional algorithms. In contrast, DGSL selects datasets based on the specific task, requiring only coarse-grained node/graph category information. This category information is typically labeled based on human-defined rules and post-hoc facts, which may change depending on the dataset or task.

- *Data Augmentation.* Since exact similarity computation in GTDGSL is often challenging, ground-truth supervision may be unavailable in some cases. In such instances, graph pairs or triplets (especially for subgraph isomorphism prediction) can be effectively generated by applying perturbations (e.g., node/edge additions, deletions, relabeling) to graphs. In contrast, for DGSL, since similarity measures in such tasks are typically unknown, generating reliable training data remains challenging. When data is scarce, techniques such as Masked Autoencoders and adversarial learning can be employed to generate additional samples. These methods simulate category distributions to generate in-distribution data, enhance data diversity.

## 3   Design Space of GTDGSL Methods

### 3.1   Overview

In this survey, we explore the design space of GTDGSL methods alongside the training pipeline, enabling alignment between model inputs and outputs and offering a clearer understanding of their operational flow. Given an *input graph pair*, GTDGSL methods typically employ deep learning techniques, such as graph neural networks, to *extract neighborhood information* for each node. This step is crucial, as it provides the basic information required for subsequent similarity computations between graph pairs. Since similarity is inherently a pairwise relationship, incorporating interactions between graphs within each pair is a natural approach to *computing similarity scores*, whether at a fine-grained or/and coarse-grained level. Building on this, the training pipeline for GTDGSL methods typically encompasses several key steps: input preparation, preprocessing, node encoding, fine-grained level scoring, graph feature generation, coarse-grained level scoring, and defining training objectives and supervision signals, as illustrated in Figure 2. Each of these steps

| | GTDGSL | DGSL | Effective Design Choices |
|---|---|---|---|
| Focus | Pairwise graph similarity | Category-level similarity | GTDGSL requires sensitivity to small structural and feature differences, while DGSL focuses on identifying features that distinguish graphs from different categories |
| Model Input | Two graphs | Single graph | GTDGSL incorporates pair-dependent features to enhance initial node/graph representations, while DGSL processes graphs independently |
| Graph Interactions | Cross-graph interactions | No direct interaction between graphs | GTDGSL applies node/graph comparisons or cross-graph fusion, balancing efficiency, effectiveness, and indexability. DGSL focuses on extracting per-graph representations |
| Model Output | Pair-dependent similarity score | Likelihood for each possible label | GTDGSL enables fine-grained similarity scoring for ranking/matching, whereas DGSL provides classification probabilities |
| Training Data | Graph pairs from diverse domains | Task-specific datasets | GTDGSL computes ground truth using conventional algorithms, while DGSL datasets are labeled based on human-defined rules or post-hoc facts |
| Data Augmentation | Perturbation-based sampling | In-distribution data generation | GTDGSL requires carefully curated synthetic data to reflect specific similarity properties, while DGSL relies on data augmentation or adversarial methods to simulate category distributions |

Table 2: Summary and Comparison between GTDGSL and DGSL

offers opportunities for divergence in model design, contributing to the expansive and diverse design space of GTDGSL methods.

In the following sections, we closely examine each step of the training pipeline shared by many GTDGSL models. We discuss common practices at each step and highlight specific optimizations introduced by various models. Additionally, at the end of each step, we highlight the key techniques and summarize their advantages and disadvantages. This structure not only highlights the evolution of GTDGSL methods but also establishes a unified framework for comparison, enabling a more systematic evaluation of their design choices and identifying key trends in the field. The categorization alongside the training pipeline, including their targeted problems and model outcomes, is summarized in Table 4.

### 3.2 Input Preparation

**Common Practice.** GTDGSL methods generally take the entire original graph pair as input and treat similarity computation as a regression/classification task. By circumventing the combinatorial search process, they enable end-to-end similarity computation, expediting the process significantly. These models aim to directly predict similarity scores for each graph pair, making them particularly advantageous in scenarios involving a large number of graphs, where fast inference and approximate similarity scores are prioritized over detailed node mappings.

**Operate in Learn-to-search Scenarios.** Certain methods treat the regression/classification task as a subcomponent, using pretrained models to evaluate partial solutions and guide the search process of conventional algorithms (Yang & Zou, 2021; Bai et al., 2021; He et al., 2022). These models process subgraphs from the original graph pair to estimate costs or scores for unprocessed parts. While subgraph inputs

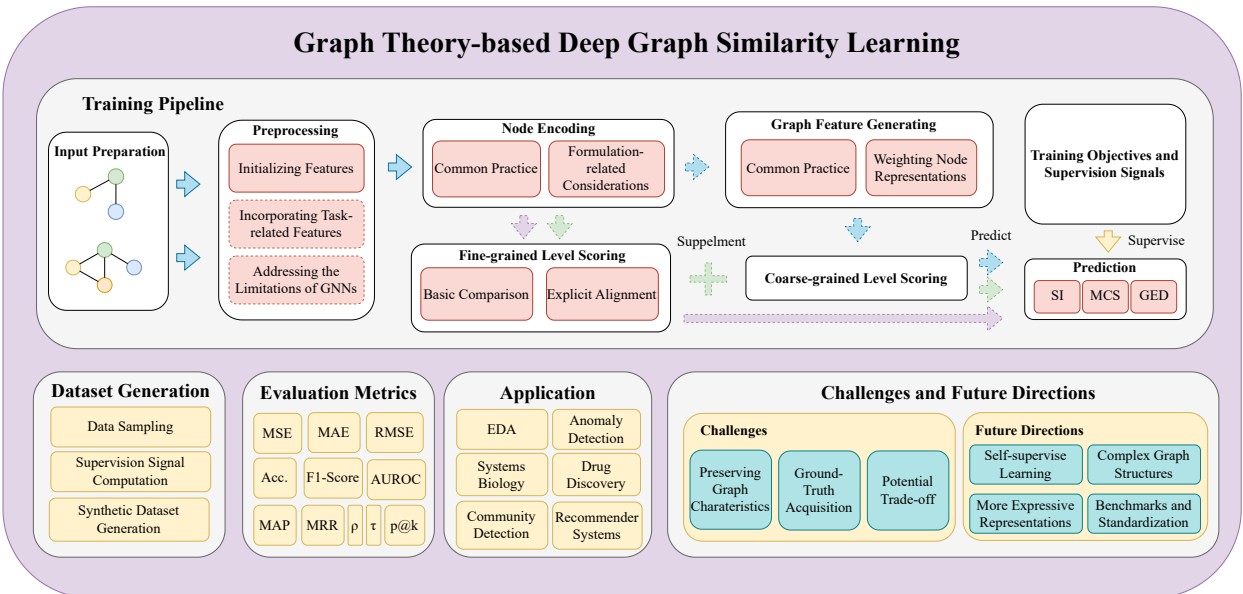

Figure 2: The visualization illustrates the organization of the survey and the training pipeline of GTDGSL methods. In the training pipeline, solid blocks or arrows represent indispensable steps or operations, while dashed ones indicate optional steps. For purple, green, or blue dashed arrows, one option must be selected based on their corresponding color.

introduce variations in the inference framework, the training pipeline remains unchanged, as the pretrained models are designed to assess similarities for both processed (i.e., subgraphs within partial solutions) and unprocessed graph parts. This survey focuses on their evaluation models, with their training objectives and supervision signals detailed in Section 3.8. For more information on the overall frameworks of learn-to-search-oriented methods, we refer readers to Yang et al. (2024).

In the following, we intentionally distinguish between the terms *substructure* and *subgraph* to avoid ambiguity. The former term refers to local structures within the input graphs, while the latter represents instances in learn-to-search scenarios where the input graphs are derived subgraphs from original graph pairs. We refer to both the original graph pair and the subgraphs derived from them as *input graphs* for simplicity.

| Methods | Basic Idea | Accuracy | Time Efficiency | Use Cases |
|---|---|---|---|---|
| Common Practice | End-to-end classification/regression | High accuracy, but depends on exact training targets | Fast inference; indexable | Graph retrieval tasks not requiring exact mappings |
| Operate in Learn-to-search Scenarios | Predict search order or the potential of partial solutions, guiding conventional algorithms | Lead to exact mappings, but may miss valid mappings; training targets may be difficult to learn | Slower due to sequential search with conventional algorithms | Tasks where a valid mapping suffices, not full enumeration, e.g., graph matching |

Table 3: Summary and Comparison Between Input Preparation Methods.

Table 4: Categorization of methods alongside the training pipeline.

| Methods | Targeted Problem | | | Input Graphs | | Scoring Level | | Model Outcomes | |
|---|---|---|---|---|---|---|---|---|---|
| | SI | MCS | GED | Original Graphs | Subgraphs | Fine-grained | Coarse-grained | Similarity | Mapping |
| SimGNN (Bai et al., 2019) | | | ✓ | ✓ | | ✓ | ✓ | ✓ | |
| GMN-emb (Li et al., 2019) | | | ✓ | ✓ | | | ✓ | ✓ | |
| GMN-match (Li et al., 2019) | | | ✓ | ✓ | | ✓ | ✓ | ✓ | |
| NeuroMatch (Ying et al., 2020) | ✓ | | | ✓ | | | ✓ | ✓ | |
| GraphSim (Bai et al., 2020) | | | ✓ | ✓ | | ✓ | | ✓ | |
| Noah (Yang & Zou, 2021) | | | ✓ | | ✓ | ✓ | | ✓ | ✓ |
| GOTSim (Doan et al., 2021) | | ✓ | ✓ | ✓ | | ✓ | | ✓ | |
| TAGSim (Bai & Zhao, 2021) | | | ✓ | ✓ | | | ✓ | ✓ | |
| GENNA* (Wang et al., 2021) | | | ✓ | | ✓ | ✓ | | ✓ | ✓ |
| GLSearch (Bai et al., 2021) | | ✓ | | | ✓ | ✓ | | | ✓ |
| H2MN (Zhang et al., 2021) | | | ✓ | ✓ | | ✓ | | ✓ | |
| EGSC (Qin et al., 2021) | | | ✓ | ✓ | | | ✓ | ✓ | |
| IsoNet (Roy et al., 2022b) | ✓ | | | ✓ | | ✓ | | ✓ | |
| RLQVO (Wang et al., 2022) | ✓ | | | | ✓ | ✓ | | | ✓ |
| Eric (Zhuo & Tan, 2022) | | | ✓ | ✓ | | ✓ | ✓ | ✓ | |
| MCSNet (Roy et al., 2022a) | | ✓ | | ✓ | | ✓ | | ✓ | |
| Prune4Sed (Liu et al., 2022) | ✓ | | | ✓ | | ✓ | | ✓ | |
| FAST (He et al., 2022) | ✓ | | | | ✓ | | ✓ | ✓ | ✓ |
| Greed (Ranjan et al., 2022) | ✓ | | | ✓ | | | ✓ | ✓ | |
| D2Match (Liu et al., 2023b) | ✓ | | | ✓ | | ✓ | ✓ | ✓ | |
| GEDGNN (Piao et al., 2023) | | | ✓ | ✓ | | ✓ | | ✓ | ✓ |
| MATA* (Liu et al., 2023a) | | | ✓ | ✓ | | ✓ | | ✓ | ✓ |
| GED-CDA (Jia et al., 2023) | | | ✓ | ✓ | | ✓ | ✓ | ✓ | |
| AEDNet (Lan et al., 2023) | ✓ | | | ✓ | | ✓ | | ✓ | |

## 3.3 Preprocessing

In existing GTDGSL methods, the primary goal of the preprocessing step is to incorporate the necessary information for the subsequent node encoding step. This involves initializing features, incorporating task-related features, and addressing the limitations of GNN.

**Initializing Features.** Before feeding input graphs into GNNs, existing GTDGSL methods generally initialize node features with either node labels (Bai et al., 2019; Li et al., 2019; Ying et al., 2020; Yang & Zou, 2021; Doan et al., 2021; Wang et al., 2021; Qin et al., 2021; Zhuo & Tan, 2022; Liu et al., 2022; He et al., 2022; Ranjan et al., 2022; Liu et al., 2023b; Piao et al., 2023) or node degree information (Wang et al., 2021; Liu et al., 2023b;a). Similarly, edge features can be initialized with edge labels (Li et al., 2019). Notably, IsoNet (Roy et al., 2022b) and MCSNet (Roy et al., 2022a), which specifically tackle feature-agnostic subgraph matching and MCS problems, initialize all node features to an identical value, ignoring the original attributes of the nodes.

**Incorporating Task-related Features.** Following conventional algorithms, some GTDGSL models incorporate task-specific knowledge through heuristics as additional node features to enhance the understanding of the problem. One notable approach is RL-QVO (Wang et al., 2022), which is designed to optimize the search order for conventional subgraph matching algorithms. RL-QVO incorporates scaled node degrees, integer labels, and query node IDs to initialize query node features, facilitating the differentiation of the input node order for the graph neural network. Furthermore, it also introduces two precomputed heuristics: (1) the frequency of vertices in the data graph $D$ with a higher degree than the query vertex $u$, and (2) the frequency of vertices in $D$ sharing the same label as $u$. These heuristics provide an estimation of the matching difficulty for each node in the query graph, helping to anticipate the potential solution space that will emerge as the matching progresses. Similarly, GLSearch (Bai et al., 2021), which operates under a branch-and-bound search framework to address the Maximum Common Subgraph (MCS) problem, leverages heuristics such as local degree profiles.

**Addressing the Limitations of GNNs.** GNNs are effective at generating graph representations, making them powerful tools for a range of graph-based tasks. However, they face limitations when dealing with complex graph structures.

- *Limited Expressiveness*: GNNs based on the message-passing mechanism focus on the local structures of nodes, producing similar representations for nodes with identical local structures. Consequently, GNNs struggle to distinguish between nodes, such as in cycle or regular graphs, where each node has the same local structure. To address this limitation, D2Match (Liu et al., 2023b) tackles the subgraph matching problem by transforming it into a subtree matching problem. It enhances performance by replacing chordless cycles shorter than length $l$ in graph pairs with supernodes, thereby avoiding ambiguities in these structures. To increase GNN expressiveness on regular graphs, NeuroMatch (Ying et al., 2020) builds on the concept introduced in (You et al., 2021). It incorporates identity information indicating whether a node is an anchor node into the features of each node-anchored subgraph, making them identity-aware and better equipped to distinguish nodes.

- *Loss of Overall Structure*: GNNs generally treat graphs as unordered sets of nodes to ensure permutation invariance, but this approach can overlook global structural information, which is crucial for targeted problems. To capture such information, MATA* (Liu et al., 2023a) perturbs graphs by randomly adding or removing edges, calculating random-walk probabilities for each node in both the original and perturbed graphs, and using these probabilities as relative position encodings. GED-CDA (Jia et al., 2023) generates spectral encodings by calculating the eigenvalues and eigenvectors, which embed the global structural properties of the graph. H2MN (Zhang et al., 2021) proposes transforming input graphs into hypergraphs to capture richer substructure information, where each hyperedge can connect multiple nodes. Specifically, H2MN performs random-walk or $k$-hop subgraph extractions around designated center nodes, viewing each extracted subgraph as a hyperedge that connects all nodes within that subgraph.

- *Lack of Scalability*: Operating GNNs on large-scale graphs requires substantial memory to store the entire adjacency matrix and node embeddings, and the burden increases with graph size and model depth. To mitigate this issue, NeuroMatch and Greed (Ranjan et al., 2022) partition large input graphs into smaller, overlapping, node-anchored substructures. By breaking down the SI problem into smaller subproblems, each substructure can be processed independently, enabling scalable parallel computations and reducing the memory and resource load. This partitioning approach helps GNNs handle large graphs more efficiently, as the localized subgraphs preserve important structural details while reducing computational demands.

## 3.4 Node Encoding

Given the input graph pairs and their features, the node encoding step involves extracting the neighboring information for each node using deep learning techniques such as GNNs.

**Common Practice.** Popular GNNs such as Graph Convolutional Network (GCN) (Bai et al., 2019; 2020; Doan et al., 2021), Graph Isomorphism Network (GIN) (Ying et al., 2020; Zhuo & Tan, 2022; Yang & Zou, 2021), Graph Attention Network (Bai et al., 2021; Liu et al., 2022; Ye et al., 2024) are commonly applied as the backbone of models. The choice of backbone in GNNs depends on how each method considers the contributions of neighboring nodes to the central nodes' representation. GCN assumes that a node's influence on its neighbors should be weighted by their degrees. It aggregates and normalizes neighborhood information using the degrees of both the central node and its neighbors, updating the central node's representation by averaging the normalized features. In contrast, GAT dynamically adjusts the contributions of neighboring nodes, assigning weights based on the similarity between the central node and its neighbors. GIN treats all neighboring nodes equally, considering them as elements of a multiset and summing their information to update the central node's representation. Another well-known GNN model, GraphSAGE, employs a sampling-based approach to aggregate information from neighbors. While this method enhances scalability, GraphSAGE is less frequently used as a backbone in GTDGSL methods. Additionally, the choice of backbone

| Methods | Basic Idea | Accuracy | Time Efficiency | Use Cases |
|---------|-----------|----------|-----------------|-----------|
| Initializing Features | Use labels, degrees, or uniform values to initialize node/edge features | Informative in general | Fast; indexable | Suitable for indexing-based applications |
| Incorporating Task-related Features | Compute features via heuristics used in classical algorithms | Task-aware; pair-sensitive | Slower; pair-dependent | On-the-fly search that requires task-specific constraints |
| Addressing the Limitations of GNNs | Add identity, structural encodings, or preprocessing to mitigate limitations of GNNs | Improves the expressiveness of initial features or scalability | Time-consuming; indexable | Boost indexing performance |

Table 5: Summary and Comparison Between Preprocessing Methods.

also depends on the data source; for instance, GENN-A* (Wang et al., 2021) utilizes SplineCNN (Fey et al., 2018) for graphs derived from 2D images. To address the potential over-smoothing issue in GNNs—where, as the depth of aggregation increases, all nodes within a graph can share a similar receptive field, thus their node representations become indistinguishable—methods, NeuroMatch (Ying et al., 2020) and Greed (Ranjan et al., 2022) introduce skip layers, which allow information from earlier layers to be directly passed to later layers and concatenate the outputs from different layers as output, to mitigate this effect.

**Formulation-related Considerations.** The choice of backbones is also related to problem formulation. To mimic the edit operations such as node/edge substitution, insertion, and deletion. TaGSim (Bai & Zhao, 2021) proposes generating type-aware graph embedding. To this end, it solely leverages the message-passing mechanism of GNN to aggregate $k$-hop node/edge-label multiset. Since H2MN (Zhang et al., 2021) transforms input graphs into hypergraphs, it encodes them with Hypergraph Convolutional Networks (HGCN), which aggregate information based on the incidence matrix of hypergraphs to generate node representations. It further devises a hyperedge pooling operation according to the Personalized PageRank (PPR) values to keep the top-ranked hyperedges based on their importance. To consider edge features, GMN (Li et al., 2019) follows the practice in Li et al. (2015) and proposes concatenating the edge information and the node information at the ends of the edge to update the nodes, which can be expressed as follows.

$$\mathbf{m}_{ji} = \text{MLP}(\mathbf{x}_i^l, \mathbf{x}_j^l, \mathbf{e}_{ij})$$

$$\mathbf{x}_i^{l+1} = \text{RNN}(\mathbf{x}_i^l, \sum_{j \in \mathcal{N}(i)} \mathbf{m}_{ji})$$

Where MLP is a Multilayer Perceptron, and RNN is a recurrent neural network, and it can be replaced by its variants such as GRU and LSTM. Such a backbone is further adopted in (Roy et al., 2022b;a).

## 3.5 Fine-grained Level Scoring

The graph similarity of input graphs can be evaluated at fine- or coarse-grained levels. Coarse-grained scoring assesses similarity by comparing embeddings representing entire input graphs, which will be discussed later. In contrast, fine-grained scoring focuses on capturing detailed structural similarities and differences at the level of nodes, edges, or substructures, and can be performed either during or after the node encoding step, with the resulting scores either supplementing or replacing the coarse-grained similarity score.

| Methods | Basic Idea | Accuracy | Time Efficiency | Use Cases |
|---|---|---|---|---|
| Common Practice | Use popular GNNs like GIN to extract node embeddings | Effective, particularly for GIN and GCN | Fast | Suitable for tasks without complex graph structures |
| Formulation-related Considerations | Select backbone based on problem-specific formulations, such as edit operations for GED or hyperedge/edge information | Better performance under specific problem setups | Slower, with added computational cost | Handling graphs with specific structural properties or problems with specific formulation |

Table 6: Summary and Comparison Between Node Encoding Methods.

Since graph structures can differ widely in node and edge counts, these methods need to handle structural irregularities. To address this and leverage GPU efficiency, they often pad the smaller graph in each pair with dummy elements. This padding ensures both graphs are represented as tensors of equal size, enabling streamlined batch processing on GPUs. The dummy elements, set to neutral values like zeros, are designed to avoid influencing similarity calculations.

Based on how they use produced scores, fine-grained scoring methods can be further categorized as follows:

- *Basic Comparisons:* These approaches rely on direct comparisons to capture the overall similarity distribution. This distribution can supplement or replace graph-level representations to predict similarity scores, as demonstrated in SimGNN (Bai et al., 2019) and GraphSim (Bai et al., 2020).

- *Explicit Alignment:* These methods establish one-to-one correspondences between nodes or edges across graphs and calculate similarity based on these mappings to minimize the overall transport cost, as exemplified by GOTSim (Doan et al., 2021).

In the following sections, we examine the fine-grained scoring techniques, highlighting their commonalities and differences to reveal emerging trends in the field.

### 3.5.1 Basic Comparison-based Models

The most representative basic comparison-based models are SimGNN (Bai et al., 2019) and GraphSim (Bai et al., 2020). Both models compare cross-graph node pairs after all or one step of node encoding and then use the pairwise inner product as similarity scores between node pairs as follows $sim(\mathbf{X}_1, \mathbf{X}_2) = \sigma(\mathbf{X}_1 \mathbf{X}_2^\top)$. To address the absence of a natural ordering among nodes within graphs, SimGNN computes a permutation-invariant but non-differentiable histogram of pairwise similarity scores, represented as $\text{hist}(s) \in \mathbb{R}^B$, where $B$ is the number of bins. This histogram serves as a supplementary feature for graph-level scores, offering insights into the scale and overall similarity distribution of the graphs. In contrast, GraphSim permutes the similarity matrix using a breadth-first search (BFS) node-ordering scheme, ensuring that nearby nodes are placed close together to capture their connections. Additionally, GraphSim treats the similarity matrices from each layer as images and employs convolutional neural networks (CNNs) to extract information from each matrix, capturing the local similarity distribution among graph pairs.

Unlike the two methods mentioned above, which compare representations after one step of node encoding, GMN-match (Li et al., 2019) scores local similarities during the node encoding process. It updates node representations in one graph based on similarity-weighted influences from nodes in the other graph. Specifically, GMN-match alternates between updating node representations within one graph and incorporating information from its counterpart. For a node $i$ in $G_1$, the model first updates its representation based on $G_1$'s local

| Methods | Basic Idea | Accuracy | Time Efficiency | Use Cases |
|---------|-----------|----------|-----------------|-----------|
| Basic Comparisons | ompute pairwise cross-graph node similarity and enhance performance with similarity distribution information | High in context prediction accuracy | Fast; highly indexable | In-context fast retrieval; may have poor generalization ability |
| Explicit Alignment | Establish implicit or explicit mapping between graphs based on node similarity | High in context prediction accuracy; improved generalization ability | Much slower; computationally intensive; indexable node-level representations | Tasks requiring interpretablity |

Table 7: Summary and Comparison Between Fine-grained Level Scoring Methods.

structure using the adjacency matrix $\mathbf{A}_{G_1}$. It then calculates the similarity between node $i$ and all nodes $i'$ in $G_2$. These similarities are used to compute attention weights $\alpha_{ii'}$, which indicate the influence of each node in $G_2$ on the updated representation of node $i$. The final representation of node $i$ is obtained by subtracting the weighted sum of representations from nodes in $G_2$, thereby integrating cross-graph information into node $i$'s feature vector. This process is formally expressed as follows.

$$\alpha_{ii'} = \frac{\exp\left(sim(\mathbf{x}_i^l, \mathbf{x}_{i'}^l)\right)}{\sum_{i' \in G_2} \exp\left(sim(\mathbf{x}_i^l, \mathbf{x}_{i'}^l)\right)}$$
$$\mathbf{x}_i^{l+1} = \mathbf{x}_i^l - \sum_{i' \in G_2} \alpha_{ii'} \mathbf{x}_{i'}^l \tag{1}$$

This approach allows the fusion of cross-graph node representation, combines both cross-graph node similarity and local graph structure, and has significantly influenced subsequent methods.

H2MN (Zhang et al., 2021) extends the practice in Equation 1 to hypergraphs. Specifically, for each hyperedge $e_i$ in a graph, it first measures its cosine similarity score with all hyperedges $e_{i'}$ in the other graph to compute cross-graph attention coefficients $\alpha_{i,i'}$ then aggregates the relevant information in the other graph based on such coefficients to compute contextual hyperedge representations. Furthermore, it compares the original hyperedge representation $\mathbf{e}_i$ with its contextual one $\tilde{\mathbf{e}}_i$ as follows $\mathbf{m}_i = \text{cosine}(\mathbf{e}_i \odot \mathbf{W}, \tilde{\mathbf{e}}_i \odot \mathbf{W})$, to compute matching vectors, which further serves as inputs for the next layers. It then concatenates the readout outputs of each graph from each layer to form the final matching representations and then predicts scores with an MLP. Prune4Sed (Liu et al., 2022) learns representations of the data graph conditioned on the query graph, generating data node embeddings that capture their potential relevance to the query. The model iteratively calculates a keep probability for each data node, pruning nodes based on these probabilities. After pruning, it computes the distance between the pruned data graph and the query graph to estimate the subgraph edit distance (SED).

Although node-, edge-, and substructure-level interactions capture fine-grained similarity between graph pairs, they have at least quadratic complexity. Eric (Zhuo & Tan, 2022) introduces an Alignment Regularization (AReg), similar to (Hassani & Khasahmadi, 2020), to reduce this complexity and avoid the need for explicit node-to-node matching. During training, AReg implicitly aligns nodes by maximizing the mutual information between node representations and the graph representations of both their own and another graph in an unsupervised manner. During inference, the learned graph-level representations are directly used to compute similarity scores, bypassing AReg to reduce the inference time.

| Methods | Basic Idea | Accuracy | Time Efficiency | Use Cases |
|---|---|---|---|---|
| SimGNN | Generate node similarity histogram features to supplement graph-level scores | Proven effective for SI, MCS, and GED tasks | Fast fine-grained scoring; node-level indexable | General GTDGSL-based retrieval |
| GraphSim | Reorders similarity matrices via BFS traversal and processes them as images using CNNs | Highly accurate in GED prediction, but convergence may be unstable and sensitive to parameter initialization | Slower; node-level indexable | GED-based retrieval |
| GMN-match | Fuses cross-graph node representations iteratively based on similarity and local structure | Highly accurate in SI, MCS and GED predictions | Slow; node-level indexable | General GTDGSL-based retrieval |
| H2MN | Extends GMN-match to fuse hyperedge representations | Proven to be highly accurate in GED prediction | Slow; complexity scales with hyperedges; node-level indexable | GED-based retrieval |
| Prune4Sed | Prune irrelevant data nodes based on the query graph | Accurate for SI prediction | Slow; node-level indexable | SI-based retrieval |
| Eric | Imposes node-graph alignment constraints during training and detaches them at inference | Highly accurate for SI, MCS, and GED; training may suffer from gradient explosion and require careful tuning | Slow training; fast inference; indexable | General GTDGSL-based retrieval |

Table 8: Summary and Comparison Between Basic Comparison-based Methods.

### 3.5.2   Explicit Alignment-based Methods

To improve the interpretability of the similarity computation process, GOTSim (Doan et al., 2021) first formulates GED computation as a graph optimal transport problem whose goal is to minimize the transport cost from one graph to another. To do so, it first proposes to explicitly establish the node correspondences between graph pairs with a linear assignment solver. Considering the characteristics of GED computation, it further proposes augmenting the cost matrix to reflect the costs of the node-deletion and the node-insertion operations, which accordingly increase the complexity of pair-wise node comparison to $\mathcal{O}(\max(|\mathcal{V}_{G_1}|, |\mathcal{V}_{G_2}|)^{2.5})$. The final graph similarity score is determined by the minimum transport cost.

IsoNet (Roy et al., 2022b) further proposes to align edges within two graphs to predict SI. Different from GED computation, SI is a partial order relationship (Ying et al., 2020), thus an edge $e$ in a query graph matches another edge $e'$ in a data graph, meaning the subgraph induced by $e$ is contained by the counterpart of $e'$. To reflect such containment constraint, the distance of two edge representations is computed as $D(e_{ij}, e_{i'j'}) = \max(0, \mathbf{e}_{ij} - \mathbf{e}_{i'j'})$, following (Ying et al., 2020). It further applied the Gumbel Sinkhorn Network to solve the optimal assignment problem, which entailed a complexity of $\mathcal{O}(k \cdot \max(|\mathcal{V}_{G_1}|, |\mathcal{V}_{G_2}|)^2)$.

Extending the practice in IsoNet, MCSNet (Roy et al., 2022a) further proposes late and early interaction variants to tackle the MCS computation problem. Using the Gumbel Sinkhorn Network, the former first computes the node embeddings of graph pairs, the latter aligns nodes during the node embedding computation step following GMN-match (Li et al., 2019). It further proposes a gossip protocol to iteratively find the largest connected component in a graph. To generate edit paths, GEDGNN (Piao et al., 2023) and MATA* (Liu et al., 2023a) adopt a similar strategy, they both propose to train a model that computes the edit paths and GED value. To this end, GEDGNN computes a matching matrix and a cost matrix of node pairs and then predicts GED based on the results of two matrices. Similarly, MATA* proposes to generate a similarity matrix of node pairs but predicts the GED based on the representations of graph pairs. Furthermore, given the $k$-best matching of the matching matrix, They both propose to compute the edit paths using conventional GED algorithms.

Orient toward SI, AEDNet (Lan et al., 2023) addresses subgraph matching by adaptively removing unnecessary edges from the data graph to better align it with the query. It follows cross-graph node representation fusion mechanism of GMN-match in Equation 1, and adapt it for SI, proposed a unidirectional cross-propagation mechanism, which transfers information from data to query nodes to align representations and approximate the ground truth matching matrix. Additionally, the model introduces a sample-wise adaptive mechanism that generates a query-specific vector, assigning minimal or zero weights to irrelevant edges. Finally, AEDNet predicts and supervises the likelihood of each node and edge in the data graph matching the query. D2Match (Liu et al., 2023b) computes an indicator matrix for graph pairs, where each entry reflects whether the subtrees rooted at nodes from the query and target graphs are subgraph isomorphic, determining SI by finding a perfect matching on a bipartite graph composed of query-data nodes.

### 3.6   Graph Feature Generation

In this step, GTDGSL methods summarize the graph representation of graph pairs using node representations, preparing for the coarse-grained level scoring. This step is not necessary for methods that solely rely on fine-grained scores.

**Common Practice.**   Existing GTDGSL methods typically generate graph-level representations using pooling techniques such as Sum (Ying et al., 2020; Bai & Zhao, 2021; Zhuo & Tan, 2022; He et al., 2022; Ranjan et al., 2022) and Max (Bai et al., 2021) pooling. Sum pooling aggregates node embeddings by performing an element-wise summation, while Max pooling selects the maximum value for each dimension across all node embeddings. These methods operate under the assumption that all nodes within the graph contribute equally to the final graph representation, without assigning different levels of importance or weights to individual nodes.

**Weighting Node Representations.**   Certain methods suggest that nodes can contribute differently to the graph representation. To consider this aspect, some methods proposed weighting node representations based

| Methods | Basic Idea | Accuracy | Time Efficiency | Use Cases |
|---|---|---|---|---|
| GOTSim | Augments node distance matrices to consider node-related edit operations, then establishes mappings using a linear assignment solver | Proven to be effective in both GED and MCS predictions | Extremely slow due to non-parallelizable solver; node-level indexable | GED/MCS-based retrieval on small graphs requiring interpretability |
| IsoNet | Computes edge distance based on a subgraph containment constraint and establishes edge mapping with the Gumbel Sinkhorn Network | Proven to be effective in SI prediction on small-sized graphs | High training and inference cost; node-level indexable | SI-based retrieval on small graphs requiring interpretability |
| MCSNet | Extends IsoNet with early/late interaction variants and enforces MCS constraints via gossip protocol | Proven to be effective in MCS prediction | Computationally intensive; node-level indexable | MCS-based retrieval on small graphs requiring interpretability |
| GEDGNN | Learns soft and hard match matrices; supervises hard match using ground-truth mappings | Proven to be effective in GED prediction | High cost in training and inference; node-level indexable | Supports GED-based retrieval for small graphs requiring interpretability; guides conventional solvers |
| MATA* | Uses the top $k$ candidates of the Gumbel Sinkhorn Network optimized mapping | Proven to be effective in GED prediction | Slow in both training and inference; node-level indexable | Supports GED-based retrieval on small graphs requiring interpretability; guides conventional solvers |
| AEDNet | Prunes redundant edges in data graph to align with query graph | Proven to be effective in SI prediction | Computationally demanding; node-level indexable | Supports SI-base retrieval on small graph requiring interpretability |
| D2Match | Converts the subgraph matching into bipartite perfect matching problem | Proven to be effective in SI prediction | High computational cost; node-level indexable | SI-based retrieval tasks requires interpretability |

Table 9: Summary and Comparison Between Explicit Alignment-based Methods.

on the node distribution of the own graph (Bai et al., 2019; Li et al., 2019). To consider the contribution of each node within a graph, SimGNN (Bai et al., 2019) proposed an attentional graph representation generation based on a global context, i.e., the average of weighted node embeddings, which can be written as follows.

$$\mathbf{c} = \tanh(\frac{1}{|\mathcal{V}|} \mathbf{W} \sum_{i=1}^{|\mathcal{V}|} \mathbf{x}_i)$$

Where $\mathbf{W} \in \mathbb{R}^{d \times d}$ is a learnable weight matrix, $\tanh(\cdot)$ is a non-linear function. Then, it considers the inner product between each node and the global context $\mathbf{c}$ to ensure greater attention for nodes similar to $c$, which can be written as follows.

$$\mathbf{G} = \sum_{i=1}^{|\mathcal{V}|} \sigma(\mathbf{x}_i^\top \mathbf{c})$$

This practice captures the compactness of node representations within graphs and focuses on nodes that are more similar to the global context. This approach is further followed by Wang et al. (2021); Zhang et al. (2021); Qin et al. (2021); Jia et al. (2023).

Similarly, GMN-emb (Li et al., 2019), the graph embedding variant of GMN-match, computes the graph representation with a gate-weighted sum of node representation:

$$\mathbf{G} = \text{MLP} \left( \sum_{i \in G} \sigma(\text{MLP}_{gate}(\mathbf{x}_i^k)) \odot \text{MLP}(\mathbf{x}_i^k) \right)$$

To handle potentially unconnected graphs, Noah (Yang & Zou, 2021) introduces a hypernode that connects all nodes for each subgraph and generates graph-level embedding through a weighted sum of node-level embeddings based on the embedding of this hypernode as follows, considering the similarity between individual node with the hypernode.

$$\mathbf{G} = \sum_{i \in \mathcal{V}_G} \sigma(\text{sim}(\mathbf{x}_i^\top, \mathbf{x}_{hyper})) \cdot \mathbf{x}_i$$

| Methods | Basic Idea | Accuracy | Time Efficiency | Use Cases |
|---|---|---|---|---|
| Common Practice | Treats all nodes equally using standard pooling operations (e.g., Sum, Max) to obtain graph representations | Robust and effective in general cases | Fast; indexable | Supports fast retrieval |
| Weighting Node Representations | Assigns learned importance weights to nodes when generating graph representations | Can outperform standard pooling; may require careful tuning | Slightly slower; indexable | Supports fast retrieval |

Table 10: Summary and Comparison Between Graph Feature Generation Methods.

## 3.7 Coarse-grained Level Scoring

In this step, GTDGSL methods compute similarity scores by comparing representations of graph pairs. In GED and MCS computations, the scores are often calculated using either the Euclidean distance between graph-level representations (Li et al., 2019) or fully connected predictors (Bai et al., 2019; Zhang et al.,

2021). Another widely used approach for scoring is the Neural Tensor Network (NTN) (Socher et al., 2013), applied in methods such as (Bai et al., 2019; Wang et al., 2021; Zhuo & Tan, 2022; Bai & Zhao, 2021; Yang & Zou, 2021; Jia et al., 2023). The NTN leverages a bilinear tensor layer to model complex interactions between graph embeddings, producing similarity scores that capture the joint influence of each embedding in the pair. Mathematically, the NTN takes two input vectors $\mathbf{e}_1$ and $\mathbf{e}_2$ and models their relationship through a bilinear tensor product:

$$sim(\mathbf{G}_1, \mathbf{G}_2) = \mathbf{G}_1^T \mathbf{W}[\mathbf{G}_2] + \mathbf{V}(\mathbf{G}_1 \oplus \mathbf{G}_2) + \mathbf{b}$$

Here, $\mathbf{W}$ is a tensor that captures the pairwise interactions in different spaces, while $\mathbf{V}$ and $\mathbf{b}$ capture linear combinations and biases. Most approaches use NTN to directly compute the similarity scores of graph pairs. In contrast, TaGSim (Bai & Zhao, 2021) considers the impact of different operations on graph structures, generating operation-specific graph representations based on node or edge representations from multiple layers. It then predicts the cost of each operation type using NTN. Supplementing this, SimGNN combines these scores with histogram features derived from pairwise node comparisons, while Eric (Zhuo & Tan, 2022) introduces a multi-scale GED discriminator that leverages NTN for interaction scoring and integrates it with the Euclidean distance between graph representations at each layer to predict.

EGSC (Qin et al., 2021) devises an attentional embedding fusion process at each layer to capture joint embeddings.

$$\mathbf{h}_{ij} = \text{CONCAT}(\mathbf{h}_i, \mathbf{h}_j)$$
$$\mathbf{h}_{ij}^* = \text{MLP}(\sigma(\mathbf{W}_U \text{ReLU}(\mathbf{W}_D \mathbf{h}_{ij})) \cdot \mathbf{h}_{ij} + \mathbf{h}_{ij})$$

The joint embeddings $\mathbf{h}_{ij}^*$ from each layer are then concatenated across layers and passed through an MLP, producing a single fused embedding to predict. To enable offline storage, EGSC introduces a method to decompose the joint embeddings into individual embeddings. This is achieved by training a student model. Specifically, the teacher model first generates self-embeddings by having each graph interact with itself, then computes pseudo-individual embeddings by subtracting these self-embeddings from the joint embeddings. The student model is trained to approximate these pseudo-individual embeddings, learning to efficiently replicate the decomposed embeddings for storage.

Unlike GED and MCS, the binary SI relationship is challenging to represent with Euclidean distance due to potential size differences between query and data graphs. To solve this issue, NeuroMatch (Ying et al., 2020) models this relationship as a partial order, predicting whether a query graph is contained by a data graph within the embedding space. This model introduces an order embedding constraint (Vendrov et al., 2016), which enforces that each dimension of the query graph's representation does not exceed the corresponding dimension in the supergraph. Violations of this dimensional ordering indicate a partial violation of the containment constraint. Greed (Ranjan et al., 2022) further extends this approach to predict both subgraph edit distance (SED). The extent of constraint violation can be computed as follows.

$$dist(\mathbf{G}_1, \mathbf{G}_2) = \sum_{i \in d} ||\max(0, \mathbf{G}_2 - \mathbf{G}_1)_i|| \tag{2}$$

Greed uses this score to approximate SED, while NeuroMatch feeds the violation score into a linear layer to predict SI. D2Match employs an NTN to compute the graph-level similarity score and combines it with an indicator matrix, which represents the subtree matching of nodes, to make predictions.

## 3.8 Training Objectives and Supervision Signals

Given the scores computed at fine-grained and/or coarse-grained levels, the training objective of GTDGSL methods typically focuses on minimizing the divergence between predicted scores and ground-truth labels. However, the way divergences are assessed and the choice of ground-truth labels, i.e., supervision signals, can vary across models.

Methods for end-to-end GED and MCS predictions, such as SimGNN, and MCSNet (Bai et al., 2019; Roy et al., 2022a), typically use Mean Squared Error (MSE) loss to quantify the error between predicted and ground-truth values, which can be either normalized or unnormalized. EGSC (Qin et al., 2021) adopts

| Methods | Basic Idea | Accuracy | Time Efficiency | Use Cases |
|---|---|---|---|---|
| Euclidean distance | Computes the Euclidean distance between graph embeddings; commonly used in GED computation | Robust and effective; provides *symmetric* scores | Fast | Supports tasks like GED under uniform cost assumptions with *symmetric* similarity |
| Fully Connected Layer | Uses a linear or multi-layer perceptron to predict similarity scores | Effective in general, but potentially suboptimal for specific tasks | Fast | General GTDGSL prediction tasks |
| Neural Tensor Network | Applies bilinear tensor transformation to model complex relationships between embeddings | Proven to be effective in GED prediction; computationally expensive and prone to overfitting | Slightly slower | Computes contextual similarity |
| EGSC | Uses attention-based fusion of graph pairs followed by an MLP for final scoring | Highly accurate in GED prediction | Fast; non-indexable graph embedding | Computes contextual similarity |
| Subgraph Containment Constraints | Enforces containment of the query within the data graph in the embedding space | Effective for SI prediction | Fast | Computes *asymmetric* similarity |

Table 11: Summary and Comparison Between Coarse-grained Level Scoring Methods.

a teacher-student pipeline for training its model on GED prediction: the teacher model minimizes GED prediction errors using MSE loss, while the student model minimizes reconstruction loss with Huber loss on pseudo-individual embeddings derived from the teacher model. To ensure the predicted similarity score falls between 0 and 1, SimGNN normalizes the score predicted by the MLP layer using a sigmoid function. In contrast, GOTSim scales the predicted distance by dividing it by the average node count of the graph pairs, accounting for varying graph sizes.

TaGSim (Bai & Zhao, 2021) introduces a slightly different training objective by predicting Graph Edit Vectors (GEV), where each entry corresponds to the cost of a specific graph edit operation. This objective provides more detailed predictions of GED but requires more sophisticated supervision using the true cost of each operation type. GEDGNN (Piao et al., 2023) supervises its model using MSE loss for the GED prediction and Binary Cross-Entropy (BCE) loss for the matching matrix, requiring the supervision of the ground-truth GED and matching matrix. Similarly, AEDNet (Lan et al., 2023), designed for SI, applies contrastive loss on the matching matrix and adaptive edge deletion. The goal is to ensure the embeddings of matched nodes are significantly more similar than unmatched nodes while simultaneously aligning the local adjacency structures by removing non-relevant edges.

In contrast, methods like NeuroMatch (Ying et al., 2020), D2Match (Liu et al., 2023b), and Greed (Ranjan et al., 2022), which address SI, make predictions based on the violation scores, which quantify the extent to which the SI relationship is violated, as outlined in Equation 2. NeuroMatch uses BCE loss to supervise the binary SI classification, along with the violation scores. It trains the model with positive examples (subgraphs of the anchor graph) and negative examples (non-subgraphs), applying a max-margin loss to ensure a clear separation between the violation scores of positive and negative pairs. D2Match, while also handling binary classification, adopts a different strategy. It uses Mean Absolute Error (MAE) loss to supervise the indicator matrix, enforcing the outputs to be either 0 (for non-matching) or 1 (for matching). Additionally, D2Match employs MSE loss to train the graph-level similarity score. On the other hand, Greed computes lower and upper bounds for SED using an approximate algorithm. The model is then trained based on these bounds using the following loss function:

$$\mathcal{L} = \max(0, \text{lb} - dist)^2 + \max(0, dist - \text{ub})^2$$

Where lb is the lower bound, ub is the upper bound, $dist$ is computed with Equation 2.

As discussed at the beginning of Section 3, approaches operating in learn-to-search scenarios propose interacting with processed and unprocessed subgraphs to capture changes introduced by the sequential decision process. Although they share a similar training pipeline with end-to-end methods, their supervision signals can differ slightly. For instance, Noah (Yang & Zou, 2021) trains its evaluation model using MSE loss, supervised by the distance between unprocessed subgraphs. GENNA* (Wang et al., 2021) supervises its model with the cost of the optimal solution under MSE loss and then fine-tunes it to predict the similarity between unprocessed graphs. GLSearch (Bai et al., 2021) uses GNNs as a DQN component to evaluate partial solutions (two subgraphs) and potential node matches (node pairs). It trains the model to predict the remaining size of the largest common subgraph starting from the current partial solutions, also using MSE loss. In contrast, RLQVO (Wang et al., 2022) adopts a reinforcement learning paradigm. It uses use the reduced number of enumeration compared with order produced by existing subgraph matching algorithm, as part of the reward signal to improve its GCN and MLP-based policy network.

## 4 Dataset Generation

**Data Sampling.** It is worth noting that since there are no specific datasets for GTDGSL tasks, existing methods often extract or sample graphs from the original datasets, such as TUDataset (Morris et al., 2020). For instance, SimGNN and GraphSim provide the pairwise GED value of graphs collected from AIDS, Linux, and IMDB-MULTI datasets[1]. They choose graphs with 10 or fewer nodes from AIDS and Linux to evaluate the efficiency, and use the full IMDB-MULTI dataset without any selection to test the scalability. NeuroMatch randomly chose an original graph $G_o$ from the dataset according to the graph scale within the

---

[1]https://pytorch-geometric.readthedocs.io/en/latest/generated/torch_geometric.datasets.GEDDataset.html

| Methods | Training Objective | Supervision Signals |
|---------|--------------------|--------------------|
| Typical GED and MCS predictions | Minimize prediction error using MSE or MAE loss | Ground-truth GED or MCS values (can be normalized to [0,1]) |
| EGSC | Distill joint embeddings via student-teacher training; minimize reconstruction error using Huber loss | Pseudo-individual embeddings derived from the teacher model |
| TaGSim | MSE loss | The edit costs for each involved operation types |
| GEDGNN | BCE loss to supervise hard matching matrix | Ground-truth mappings |
| AEDNet | Contrastive loss to supervise matching matrix and edge deletion | Ground-truth mappings |
| NeuroMatch | BCE loss for binary SI prediction; max-margin loss for containment violation | SI relationship |
| D2Match | MAE loss for matching indicator; MSE loss for graph-level similarity | SI relationship |
| Greed | Dual max-margin loss on violation scores or distances | Upper and lower bounds of SED or GED |
| Noah | MSE loss | Estimated GED between unprocessed subgraphs |
| GLsearch | MSE loss | Remaining size of the largest common subgraph from current partial solution |
| RLQVO | Policy reward loss | Reduction in enumeration compared to traditional search order |

Table 12: Summary and Comparison Between Training Objectives and Supervision Signals.

dataset. Based on the chosen graph, it samples the anchor graph by randomly choosing a central node $u$ and performing a random breadth-first traversal (BFS) of the graph, extracting the traversed substructure, and sampling its positive examples by performing the same process on the anchor graph starting from the same central node. Then, it proposes to sample negative examples from $G_o$ starting from a node other than $u$ or perturb the sampled positive examples to make it no longer a subgraph of the anchor graph.

**Supervision Signal Computation.**   To generate ground-truth supervision signals, conventional exact or approximate algorithms are often used. For example, exact GED can be computed with the A* algorithm (Abu-Aisheh et al., 2015)[2] and more advanced A*LSa (Chang et al., 2020)[3], though A*LSa does not support customizable edit costs. Alternatively, the smallest distance among Beam, Hungarian, and VJ algorithms can serve as an approximation[4]. GEDLIB (Blumenthal et al., 2019)[5] offers another approximate method that computes lower and upper bounds for GED and SED, as used in (Ranjan et al., 2022). The exact

---

[2] https://networkx.org/documentation/stable/reference/algorithms/generated/networkx.algorithms.similarity.graph_edit_distance.html
[3] https://github.com/LijunChang/Graph_Edit_Distance
[4] https://github.com/dzambon/graph-matching-toolkit
[5] https://dbblumenthal.github.io/gedlib/

MCS value can be computed with MCSplit (McCreesh et al., 2017) [6]. The algorithm for the exact subgraph isomorphism computation[7, 8], including VF3 (Carletti et al., 2018) and other algorithms, as examined in (Sun & Luo, 2020). RLQVO evaluates its ordering planning for subgraph matching using datasets from previous studies (Sun & Luo, 2020).

**Synthetic Dataset Generation.** Since computing ground-truth solutions for this graph theory problem is intractable, synthetic GED datasets have become a promising alternative. In this context, GMN generates training data by sampling random binomial graphs with a specified number of nodes and edge probability. From each synthetic graph, it creates positive and negative examples by randomly substituting edges, ensuring a greater number of substitutions for the negative examples than for the positive ones. As exact GED methods provide only an overall GED score for a graph pair without tracking fine-grained values for each graph edit type in GEV, TaGSim proposes generating synthetic graph pairs that adhere to a specified GEV. Meanwhile, GLSearch validates its performance on large graphs by sampling a connected subgraph twice to produce two overlapping subgraphs. NeuroMatch, on the other hand, generates Erdos-Rényi (ER) random graphs and extended Barabasi (BA) graphs as base graphs and then applies a BFS strategy to generate positive and negative examples. To further investigate model generalization, NeuroMatch suggests sampling unseen queries from various distributions, including random BFS, degree-weighted sampling, and random walk sampling.

## 5 Evaluation Metrics

Since GED and MCS similarity computations are generally formulated as regression tasks. The GTDGSL models for GED and MCS are often evaluated by Mean Squared Error (MSE), Mean Absolute Error (MAE), or Rooted Mean Squared Error (RMSE) to quantify the gap between the predicted value and the ground truth. Towards scenarios such as graph search for a database, which requires returning the top-$k$ similar data graph for a given query graph, the ranking metrics such as Spearman's Rank Correlation Coefficient ($\rho$), Kendall's Rank Correlation Coefficient ($\tau$) and Precision at k ($p@k$) are also applied.

In addition to the above metrics, learn-to-search-oriented models, such as Noah, also propose the use of accuracy and feasibility. The former measures the accuracy of the computed GEDs compared to the ground-truth GEDs, the latter measures the ratio that the computed GEDs are feasible (i.e., they are equal to or smaller than the ground-truth GEDs). In contrast, metrics such as Area Under the Receiver Operating Characteristic Curve (AUROC), Accuracy, and F1-score are adopted in evaluating subgraph isomorphism prediction, and Mean Average Precision (MAP) and Mean Reciprocal Rank (MRR) can be further adopted to evaluate the ranking ability, as in (Roy et al., 2022b).

## 6 Applications

Graphs are ubiquitous in numerous fields, serving as essential structures for representing complex relationships and interactions. Before the advent of GTDGSL methods, SI, MCS, and GED computations were already extensively applied across various domains to measure graph similarity. They differ in their problem formulations and properties. Each of them have unique strengths and limitations.

**Circuit Design.** GTDGSL methods can be applied in circuit design (Ohlrich et al., 1993; Lu & Pingali, 2018; Shrestha & Savidis, 2024; Li et al., 2024), where circuits are modeled as graphs, with nodes representing components (e.g., transistors or logic gates) and edges representing connections. One possible application is transistor-to-gate netlist conversion, which identifies standard logic gates (e.g., AND, OR, NAND) within a transistor-level circuit. In this context, the query graph represents a known subcircuit, and the target graph is the full transistor netlist. The goal is to find instances of the query graph within the target graph, allowing for the replacement of recognized subcircuits with their corresponding logic gates.

---

[6] https://github.com/jamestrimble/ijcai2017-partitioning-common-subgraph/tree/master
[7] https://github.com/RapidsAtHKUST/SubgraphMatching
[8] https://github.com/MiviaLab/vf3lib

SI models are well-suited for exact subcircuit identification but may encounter limitations when small modifications or alternative implementations are present. In these cases, MCS methods can detect the largest common structure between circuits, even with minor differences, enabling the recognition of functionally similar but differently implemented subcircuits. Additionally, GED quantifies structural differences, such as gate substitution, wire rerouting, or redundant logic, which is useful for technology-independent recognition.

**Cheminformatics.** In cheminformatics, a molecule is typically represented as a graph, with nodes representing atoms and edges representing chemical bonds. In these graphs, node labels indicate atom types (e.g., carbon, oxygen, nitrogen), and edge labels denote bond types (e.g., single, double, aromatic). Given a query molecular graph, the goal is to identify molecules with similar structures within a chemical database. This is crucial for drug discovery (Mohamed et al., 2019; Jayaraj et al., 2016; Shiokawa et al., 2024; Naoi & Shiokawa, 2023; Ranu & Singh, 2012; Schadt et al., 2009), as similar compounds may exhibit similar biological activity.

In this context, SI models can be used to detect the presence of functional groups or core structures within a molecule. For example, if a molecule contains a known pharmacophore (a substructure responsible for its biological activity), SI can confirm its presence. However, SI does not provide a measure of molecular similarity beyond exact substructure matching. MCS is effective for identifying structurally similar compounds with slight modifications, while GED offers greater flexibility, providing a graded measure of molecular similarity based on chemical transformations.

**Social Network Analysis.** GTDGSL methods are valuable in social network analysis (Guo et al., 2022), particularly for community detection, where they identify clusters within networks by examining structural similarities between user profiles or groups. This capability is useful for detecting anomalous patterns, such as fraudulent activities in social networks or online platforms (Xiang et al., 2009; Cui et al., 2014; Rong et al., 2018; Sangkaran et al., 2020). In these networks, nodes represent user profiles or accounts, and edges represent interactions between users, such as friendships, direct messages, or posts. Fraudulent behaviors, including botnet activity or click-farming, often exhibit structural differences from typical user interactions. For instance, while users on an online platform usually interact within small friend groups, fake accounts may engage with large, unrelated user sets, causing significant deviations in interaction patterns.

GTDGSL models can cluster users based on shared interaction patterns (e.g., content type) and detect isolated groups with abnormal patterns. SI models can detect exact matches to known fraud patterns but may be less effective at identifying evolving or novel fraud schemes. MCS helps identify commonalities between groups but may struggle with sparse or loosely connected fraudulent networks. GED can quantify the structural differences between normal and suspicious networks, although it may lack the granularity needed to capture subgraph-level anomalies critical for fraud detection in dynamic social networks, necessitating further processing.

**Recommender Systems.** In e-commerce and streaming services, GTDGSL methods can enhance item recommender systems (Lalithsena et al., 2016; Wang et al., 2023). These applications often employ graph-based approaches to model relationships between users, items, and their interactions (e.g., purchases, clicks, ratings). Given a new user or item, the goal is to identify the most similar existing users or items to provide recommendations. SI can be used to find users with identical interaction patterns. If a new user's interaction subgraph exactly matches that of an existing user, we can assume they have the same preferences and recommend the same items. However, exact matches are rare in real-world scenarios. MCS can identify users with highly overlapping preferences, even when some interactions differ. For example, if two users share most of their past purchases but have slight variations, MCS captures this similarity while allowing for differences. GED provides a quantitative measure of user similarity by computing the minimum edit cost needed to make two users' interaction graphs identical. This is particularly useful for handling sparse data, where users may have few interactions but still exhibit similar preferences.

**Protein Interaction Networks.** In systems biology, GTDGSL methods can be applied to analyze protein interaction networks (PINs) and identify potential interactions based on graph similarities, thereby contributing to the understanding of disease mechanisms (Koch et al., 1996; Peng & Tsay, 2010; Shen et al.,

2012; Ibragimov et al., 2013; 2014). In these networks, nodes represent proteins within a biological organism, while edges represent interactions between proteins, such as binding events. Researchers may compare two protein interaction networks, for example, one derived from healthy cells (the query graph) and another from cancer cells (the data graph). In this context, SI could be used to detect exact matching subgraphs representing well-known disease-associated protein complexes. MCS, on the other hand, could identify the most significant common interactions, revealing critical connections shared across multiple species or conditions. GED could be used to measure differences between healthy and cancerous PINs, helping to identify proteins whose interactions are disrupted in cancer.

**Mining Pipeline.** The integration of SI, MCS, and GED into a mining and filtering pipeline provides a comprehensive graph similarity measure, addressing both exact and fuzzy matching. This pipeline is applicable in various domains, such as circuit design, where it efficiently process a large amount of target circuits, identifies and verifies subcircuit patterns. In the pipeline, MCS and GED are initially used to mine similar subcircuit patterns from circuits with known properties. For example, when extracting potential Trojan structures from a circuit known to contain a Trojan, MCS identifies functionally similar subcircuits, while GED quantifies the structural differences between these subcircuits and the target structure. Once the patterns are mined, SI is applied for exact matching within the target circuit. If the occurrence rate of the patterns does not exceed a certain threshold, MCS and GED can be iteratively applied to refine the patterns until the rate meets a certain threshold.

# 7 Challenges and Future Directions

## 7.1 Challenges in GTDGSL

Given the detailed analysis of existing models, in this section, we discuss the general challenges or open problems encountered by GTDGSL methods.

### 7.1.1 Preserving Graph Characteristics

**Scale Information.** Scale information is crucial in similarity assessment because it impacts how graph differences are interpreted, especially when comparing graphs of different sizes. In MCS and GED computations, the raw measures tend to be biased towards larger graphs, as they naturally have more nodes and edges, making them more likely to have a higher commonality (in MCS) or a larger number of required edits (in GED). This bias can distort the similarity score, making larger graphs seem more similar to or different from each other than they actually are. Therefore, considering the scale information helps to compare graphs of varying sizes in a fair and proportional way. Scale information is also essential in SI, which aims to determine whether a substructure in one graph exactly matches another graph. This inherently involves considering the relative sizes of the graphs. Despite the importance of scale information for MCS, GED, and SI, each of them encounters challenges related to handling scale in different ways.

- *Loss of Scale Information.* In the context of graph similarity, GNNs primarily focus on local structural similarity, often overlooking the sizes of graphs. They project nodes with varying local neighborhood sizes onto a single point within the embedding space, potentially losing scale information. SimGNN (Bai et al., 2019) addresses this by summing unnormalized weighted node representations to reflect graph size. Yet, since node representations are a single point, it is unclear whether this method fully captures scale information. SimGNN also generates histogram features based on pairwise node similarity. Intuitively, the height and distribution of bins can depict the scale information and overall similarity. However, SimGNN proposes to normalize the histogram, potentially causing bins with different counts to have the same proportion. This normalization can obscure the differences in graph size, diminishing the role of scale in the final similarity assessment. Moreover, the histogram feature is non-differentiable, meaning it cannot be optimized through backpropagation. Similarly, GOTSim (Doan et al., 2021) proposes normalizing the total transport cost by the average node size of graph pairs. This may limit the model's ability to learn from the data and embody the inductive biases that designers intend to introduce.

- *Scale Difference.* Unlike MCS and GED, SI is a binary classification task rather than a numerical similarity measure. The scale difference between the query and data graphs introduces a unique challenge, as it is often difficult for GNNs to ensure that node representations for matchable pairs are similar when their neighborhoods differ greatly in size. This scale disparity can make it hard to correctly identify subgraph matches. Different methods approach this challenge in various ways. NeuroMatch (Ying et al., 2020) leverages the size difference between the query and data graphs as a filtering mechanism, using order embeddings to ensure that the representation of a data graph contains that of its subgraphs. However, because larger graphs inherently have more complex structures and a broader receptive field during GNN's message-passing process, their representations can inadvertently encompass smaller graphs', making it difficult to maintain strict containment constraints. AEDNet (Lan et al., 2023) proposes a different strategy, pruning irrelevant edges from the data graphs to reduce the representation difference between matched node pairs. However, this method may be unstable, as it heavily relies on the correct identification and removal of redundant edges. The model's ability to capture the precise subgraph structure may be compromised if crucial edges are mistakenly removed.

**Structural Information.** Capturing Structural Information that depicts the node connections within graphs is crucial for MCS, GED, and SI computations, as these problems focus not just on the local structure around nodes but also on the nodes' positions relative to each other. Since graphs have varying sizes and, within which nodes do not have a natural order. Thus, nodes within graphs generally are treated as a bag of elements during the message-passing and pooling process to emphasize the permutation-invariant nature of graphs. This practice, although shown to be effective in most applications, can cause the loss of structural information, leading to inaccurate predictions. Furthermore, within the embedding space, nodes that share similar local substructures can end up with similar embeddings. Based on this inductive bias, two different nodes in a graph that share isomorphic local substructures can have an identical representation and, thus, cannot be distinguished, which is termed automorphism in (Chamberlain et al., 2023), further confusing the matching process. Despite approaches that operate in learn-to-search scenarios employing a sequential searching process, the loss of structural information can still impede cost estimation for unprocessed subgraphs, leading to a diminished boost in search efficiency.

- *Capture Structural Relationships.* To capture the structural relationships between nodes, methods such as Laplacian matrix-based spectral encoding (Jia et al., 2023) and BFS-based ordering schemes on cost matrices (Bai et al., 2020) have been proposed. The spectral encoding utilizes the eigenvalues and eigenvectors of the Laplacian matrix to offer insights into node connections and overall graph structure. In contrast, the BFS-based ordering arranges the cost matrix based on a breadth-first search traversal, aiming to align similar structures between graphs more effectively during comparison. However, both methods have notable limitations. Spectral encoding often struggles to differentiate between automorphic nodes, as it can yield similar spectral embeddings for such nodes, hindering its effectiveness in accurately distinguishing them. On the other hand, the BFS ordering captures only 2-hop local connectivity when encountering complex graphs. Additionally, both methods are computationally demanding, with complexities of $\mathcal{O}(|\mathcal{V}|^3)$ for spectral encoding and $\mathcal{O}(|\mathcal{V}|^2)$ for BFS ordering scheme in their worst cases.

- *Break Automorphism.* To break automorphism, identities can be assigned to each node within the graph as augmented features, such as unique numbers (Wang et al., 2022) or random-walk-based features (Liu et al., 2023a). However, while unique numbers are deterministic, they do not generalize well across different graphs and may lead to a loss of permutation invariance. Conversely, random-walk-based features can be computationally expensive, especially in large graphs, as they often require multiple walks or sampling processes. Additionally, being probabilistic, random-walk-based features can produce slightly different representations for the same graph across different runs, potentially affecting their consistency. To enhance the stability of random-walk-based features, MATA* (Liu et al., 2023a) suggests augmenting graphs by randomly adding or removing edges and conducting random walks on both the original and augmented graphs, which further increases computational costs.

### 7.1.2 Ground-Truth Acquisition

The problems of SI, MCS, and GED are notoriously challenging due to their combinatorial nature, which complicates the acquisition of ground truth. Recent research indicates that computing the exact GED is particularly difficult, even for graphs with as few as 16 nodes (Blumenthal & Gamper, 2020). For datasets with more than ten nodes, ground-truth GED values are typically derived from the smallest distances calculated by approximate algorithms. However, studies (Bai et al., 2019; 2020; Doan et al., 2021; Zhuo & Tan, 2022) suggest that these approximate algorithms often perform significantly worse than current neural network methods in exact GED prediction, raising concerns about the reliability of the approximate GED values they provide. Since learning-based methods rely on GED values for training and supervision, models trained on these approximate values may inherit biases and struggle to compute GED accurately. Compared with GED computation, MCS algorithms can handle somewhat larger graphs, but their scalability remains limited. The most advanced MCS algorithm, MCSplit (McCreesh et al., 2017), can solve 2,000 out of 4,110 unlabelled, undirected instances with up to 50 nodes per graph, given a time limit of 0.5 seconds per instance, as shown in the original paper.

In contrast, subgraph matching algorithms can process significantly larger graphs. State-of-the-art subgraph matching frameworks, such as RapidMatch (Sun et al., 2020), can enumerate the first $10^5$ mappings for queries with 32 nodes in large-scale graphs, such as YouTube, which contains 1,134,890 nodes and 2,987,624 edges, within 100 seconds. However, it is important to note that subgraph matching algorithms aim to enumerate mappings for a given query, whereas learning-based SI predictors focus on predicting the existence of a SI relationship. In cases of positive instances, where a SI exists, subgraph matching algorithms can terminate early upon finding a valid mapping. Conversely, for negative instances, where no SI exists, these algorithms may have to exhaustively search the solution space to confirm the absence of any matching subgraph. This exhaustive search can be more time-consuming than identifying a positive match, particularly as the size and complexity of the graph increase. While numerous large-scale datasets are available for subgraph matching, there is a notable lack of datasets specifically designed for SI prediction.

Although some methods (Li et al., 2019; Bai & Zhao, 2021; Piao et al., 2023; Ying et al., 2020) propose training and testing models on generated datasets for MCS, GED, and SI, these datasets are often created by modifying original graphs to meet target prediction values. Such modifications require careful design, or the target prediction values given during the generation process may not represent the optimal solution for the graph pairs. This is because modifications made to a subgraph extracted from a larger graph may inadvertently make it resemble other parts of the larger graph.

Recent research toward search-to-learn scenarios has shown that conventional algorithms empowered by deep learning techniques can deliver better solutions on larger datasets in less time compared with the original ones (Liu et al., 2023a; Bai et al., 2021). However, due to the approximate nature of neural network-based approaches, they may overlook critical information, potentially missing valid solutions and compromising the approach's overall reliability (He et al., 2024).

### 7.1.3 Potential Trade-off Between Performance and Scalability

The trade-off between performance and scalability poses significant challenges in the GTDGSL problem. Notably, these aspects are rarely explored in existing studies, with the exception of Piao et al. (2023), which evaluates the performance of methods such as Noah (Yang & Zou, 2021), a model that predicts costs based on graph representations, and GEDGNN (Piao et al., 2023), which leverages both a cost matrix and a matching matrix for cost prediction. The results indicate that the performance of both methods declines as graph size increases, particularly for Noah. This suggests that while methods predicting using graph representations are generally more efficient than those that implicitly or explicitly establish cross-graph node mappings, they may generalize worse on larger graphs. Although explicit alignment-based methods such as GEDGNN may perform better on large graphs, they typically have at least quadratic complexity, which may hinder the ability to handle large graphs.

## 7.2 Future Directions

Based on the above limitations, several promising directions for future research can be identified:

**Self-Supervised Learning.** The integration of self-supervised learning offers a promising solution to the challenges of data acquisition in GTDGSL methods. By leveraging abundant real-world graph pairs without ground-truth labels, self-supervised techniques can reduce reliance on computationally expensive datasets for GED, MCS, and SI. Exploring diverse self-supervised objectives, such as contrastive learning or masked graph prediction, could enhance the adaptability and performance of GTDGSL methods, making them more suitable for real-world applications.

**Handling Complex Graph Structures.** Currently, most GTDGSL methods are limited to undirected graphs without edge labels, which restricts their utility across broader graph types. Extending these models to support directed graphs with labeled edges would enable them to better capture the nuanced, directional relationships that often exist in real-world scenarios such as knowledge graphs. Furthermore, expanding GTDGSL methods to handle heterogeneous graphs—those with multiple types of nodes and edges—presents an important research direction. Heterogeneous graphs require specialized representations and similarity metrics that account for diverse entities and relationships. Developing these capabilities could significantly enhance the applicability of GTDGSL methods in areas such as social networks, biomedical research, and recommender systems, where accurate similarity assessments are essential in multi-relational and data-rich environments.

**More Expressive Representations.** Another key direction is advancing graph representations to enhance GTDGSL methods. This involves overcoming the limitations of the 1-WL test, increasing sensitivity to global graph structures, addressing computational constraints, and mitigating over-smoothing in GNNs. These improvements could enable more effective capture of fine-grained differences between graphs.

**Benchmarks and Standardization.** Establishing benchmarks and standardizing evaluations are essential for advancing GTDGSL research. The lack of standardized datasets that cover diverse graph sizes and are tailored to GTDGSL challenges hinders consistent assessment and comparison of methods. Developing benchmark datasets and tasks that reflect real-world applications would improve reproducibility, foster collaboration, and enable the identification of best practices, ultimately driving progress in graph similarity learning.

## 8 Conclusion

This survey presents a comprehensive overview of Graph Theory-based Deep Graph Similarity Learning (GTDGSL). To the best of our knowledge, it is the first work to examine graph similarity learning methods based on graph theory concepts, including subgraph isomorphism, maximum common subgraph, and graph edit distance. We review existing GTDGSL methods, analyzing their training pipelines and techniques to identify commonalities and distinctions. Through this analysis, we highlight current technical trends, applications, and key challenges of GTDGSL methods. Despite promising results in achieving interpretable graph similarity with high efficiency and accuracy, GTDGSL methods face key limitations, such as challenges in preserving critical graph characteristics and obtaining ground-truth supervision signals. These limitations highlight future directions, including the development of more expressive representations and the adoption of self-supervised learning approaches.

## Acknowledgments

This work is supported in part by National Key Research and Development Program of China (No. 2023YFB4502300) and the National Natural Science Foundation of China under grants (Nos. 62402503, 62025208 and 62421002).

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
