# OpenReview forum: "Graph Theory-Based Deep Graph Similarity Learning: A Unified Survey of Pipeline, Techniques, and Challenges"
_TMLR — Accepted by TMLR_

### Review · Reviewer_ofZ6 · 2025-01-11

**Summary Of Contributions:**

This survey paper provides a comprehensive overview of graph theory-based deep graph similarity learning, focusing on integrating graph theory concepts into deep learning frameworks for measuring graph similarity.

**Audience:**

Yes

**Claims And Evidence:**

Yes

**Requested Changes:**

See Cons.

**Strengths And Weaknesses:**

Pros:
1. The paper offers a thorough review of GTDGSL methods, making it a valuable resource. 2. It is well-organized and easy to follow, with a unified framework for method comparison. 3. It provides useful insights into the current trends and future directions in the field.

Cons:
1.  The paper reviews numerous methods but often lacks a critical analysis of their limitations. For example, the discussion on GNNs could include their known issues like over-smoothing and challenges with very large graphs. A more balanced view would be beneficial. 2. The paper mentions applications but lacks detailed case studies. Including specific examples of how these methods have been applied in real-world scenarios would greatly enhance the practical relevance of the survey. 3. The paper discusses subgraph isomorphism, maximum common subgraph, and graph edit distance separately but does not explore how they can be integrated. A discussion on combining these concepts for a more comprehensive measure of graph similarity would be valuable. 4. A relevant survey that seems to be missing is "A Comprehensive Survey on Deep Graph Representation Learning." 5. The paper focuses on deep learning methods but does not provide a thorough comparison with traditional non-deep learning methods.

---

> ### Author Response · Authors · 2025-02-12
> **Response Part 1**
>
> Dear Reviewer ofZ6,
>
> Thank you for your detailed review and valuable feedback. We appreciate the time and effort you invested in evaluating our work. We have carefully considered your suggestions and made the following **revisions**, which are **highlighted in orange within the revised manuscript**, to address the concerns you raised:
>
> - **Con 1**: The paper reviews several methods but lacks a critical analysis of their limitations, particularly regarding GNNs' known issues like over-smoothing and challenges with large graphs.
>
> - **Response**: We greatly appreciate this insightful comment. In response to the reviewer's feedback, we have revised **Section 2.2.1** to provide an analysis of key GNN limitations, including **over-smoothing, 1-WL test-bounded expressiveness, loss of overall structural information, and scalability issues with large graphs**, along with how mainstream research addresses these challenges.
>
>   Beyond this section, we have already discussed how GTDGSL approaches mitigate these issues:
>
>   - **Over-smoothing** is covered in **Section 3.4 "Node Encoding"-"Common Practice"**.
>   - **Other GNN limitations** are analyzed in **Section 3.3 "Addressing the Limitations of GNNs"**.
>   - **Challenges crucial to GTDGSL**, such as preserving **scale** and **structural information**, are examined in **Section 7.1.1 "Preserving Graph Characteristics"**.
>
>   To further guide readers, **Section 2.2.1** now includes references to relevant literature and explicitly points to later sections for GTDGSL solutions. We hope these revisions sufficiently address your concerns.
>
> - **Con 2**: While the paper mentions some applications, it lacks specific case studies.
>
> - **Response**: In response to the reviewer’s suggestion, we have revised **Section 6 "Applications"** to provide a more structured presentation of the applications of GTDGSL. We have added brief case studies for each application area, detailing how graph representations are defined, what constitutes the query and target graphs for each specific task, and the corresponding use cases for each concept.
>
> - **Con 3**: The paper discusses subgraph isomorphism, maximum common subgraph, and graph edit distance separately but does not explore how they could be integrated for a more comprehensive similarity measure.
>
> - **Response**: We appreciate the reviewer’s insightful suggestion. In response, we have added an application case study, **"Mining Pipeline"**, in **Section 6 "Applications"** to illustrate how subgraph isomorphism (SI), maximum common subgraph (MCS), and graph edit distance (GED) can be integrated for a more comprehensive similarity measure. Specifically, MCS and GED can be used to mine similar patterns from data graphs with known properties, while SI serves as a verification step to confirm the presence of mined patterns in data graphs. If the occurrence rate of the patterns does not exceed a certain threshold, MCS and GED can be iteratively applied to refine the patterns until the threshold is met.
>
> - **Con 4**: A relevant survey, "A Comprehensive Survey on Deep Graph Representation Learning," seems to be missing.
>
> - **Response**: Thank you for bringing this to our attention. The mentioned survey [1] provides valuable insights into deep graph representation learning, focusing on general GNN architectures, learning paradigms, and applications. In contrast, our work specifically examines graph theory-based deep graph similarity learning (GTDGSL), which approximates three selected graph theory concepts for quantifying graph similarity and provides an in-depth review of GTDGSL-related techniques.
>
>   While both fields leverage GNNs and share certain challenges arising from GNNs, GTDGSL is explicitly constrained by graph theory properties, which we have added discussion in **Section 2.1 "Graph Similarity Concepts in Graph Theory"** and **Section 2.4 "GTDGSL vs. Deep Graph Similarity Learning (DGSL),"** thus leading to distinct challenges and solutions. For example, in SI prediction, a model must respect partial order relationships between subgraph isomorphisms, a constraint not present in general graph representation learning.
>
>   We believe our contributions complement rather than overlap with the mentioned survey. To acknowledge its relevance, we have now referenced it in the updated **Introduction** section.

---

> ### Author Response · Authors · 2025-02-12
> **Reponse Part 2**
>
> - **Con 5**: The paper focuses on deep learning methods but does not provide a thorough comparison with traditional, non-deep learning methods.
>
> - **Response**: We agree with your observation. To address this, we have expanded the discussion to include a detailed comparison between deep learning approaches and conventional algorithms in **Section 2.3 GTDGSL vs. Conventional Algorithms**. We compare conventional algorithms and GTDGSL across several key aspects, such as Correctness, Efficiency and Scalability, Generalization Ability,  Intermediate Result Reusability, as well as Training and Inference Time. We hope this comparison allows readers to better understand the strengths and weaknesses of each type approach, and when one might be more suitable than the other.
>
> ------
> We sincerely thank you for your constructive feedback. We believe the revisions made in response to your suggestions have significantly improved the quality and depth of the paper. We look forward to your further evaluation.
>
> [1] Wei Ju, Zheng Fang, Yiyang Gu, Zequn Liu, Qingqing Long, Ziyue Qiao, Yifang Qin, Jianhao Shen, Fang Sun, Zhiping Xiao, et al. A comprehensive survey on deep graph representation learning. Neural Networks, pp. 106207, 2024.

---

> ### Author Response · Authors · 2025-02-18
>
> Dear Reviewer ofZ6,
>
> Once again, we sincerely appreciate your detailed review and valuable feedback. As the discussion period concludes, we would like to confirm whether our responses align with your comments and if the revisions meet your expectations. Please let us know if you have any further suggestions. We would be happy to address any remaining concerns.

---

### Review · Reviewer_dMz7 · 2025-01-31

**Summary Of Contributions:**

The current paper is a survey on techniqus that quantify similarity between graphs, focusing specficially on methodologies derived from graph theory.
Starting from a brief overview on graph similarity techniques and a description of the advantages of similarities based on graph theory, the authors move on to clarify their difference with previous surveys and the overall contributions.

Section 2 describes each of the three graph similarity techniques that the paper is focusing on, specifically subgraph isomorphism, maximum common subgraph and graph edit distance,  along with a general explanation of how graph neural networks learn to perform these techniques.

Section 3 provids a detailed breakdown of the training modeling choices.
The preprocessing step is separated into the list of node features commonly used and the possible graph changes utilized to account for GNN limitations such as scalability.

The encoding step delineats the neural graph architectures utilized to learn the metrics, most of them corresponding to the GNNs standard.

The fine-grained section adresses the methods that measure structural similarities in smaller levels then the graph, which can either be supporting features for the coars-grained similarity on the graph level or act as standalone similarity measures by aggregations.

The graph feature generation explains the graph pooling methods that are common between tasks.

The graph embedding step focuses on the equations used to compute the dimilarity between the representations derived from the previous steps. Finally the training objectives covers the cost functions

The final sections go through the datasets, evaluation metrics, common applications, and open challenges for deep graph similarity models.

**Audience:**

Yes

**Broader Impact Concerns:**

There is no clear ethical concern from the paper.

**Claims And Evidence:**

Yes

**Requested Changes:**

In section 3.4 the message passing of GMN (Li et al.) is chosen to be shown in equations. It is unclear why this model stands out from the rest which are not described.
Ideally, the authors could add one table per similarity method, with the message passing, cost function, and other properties for each learning model to facilitate a thorough and fast comparison.

Since the survey focuses only on three core methodologies, it can be more 'educational' i.e. by adding figures that juxtapose them with
toy examples.

Similarly, in section 2.2.2,  it could be meaningful to show the formulation of a bare-minimum model and how it changes to learn each of the metrics.

Moreover, although the technical details are addressed adequately overall in sections 3.3-3.8, it is hard to follow which method pairs with which similarity technique (which to my understanding is important to get an overall sense of how the methods compare, based on the paper's introduction). A specific label or color could be helpful for this.

Language is redundant in some cases e.g. section 4 first and second paragraphs overlap.

**Strengths And Weaknesses:**

Overall, the paper has substantial coverage of the literature.

The taxonomy is meaningful and quite clear.

The technical aspects and differences between models are explained adequately for the most part in section 3.

---

> ### Author Response · Authors · 2025-02-12
> **Response Part 1**
>
> Dear Reviewer dMz7,
>
> Thank you for your thoughtful and constructive feedback on our paper. We appreciate the time you took to review the manuscript, and we have carefully considered your suggestions. Below, we provide a detailed response to each of the points you raised, along with **the corresponding changes we have made**, which are **highlighted in magenta within the revised manuscript**:
>
> - **Requested Change 1:** In section 3.4 the message passing of GMN (Li et al.) is chosen to be shown in equations. It is unclear why this model stands out from the rest which are not described. Ideally, the authors could add one table per similarity method, with the message passing, cost function, and other properties for each learning model to facilitate a thorough and fast comparison.
>
> - **Response:** We presented GMN [1] in equations because it employs a cross-graph node representation fusion mechanism to propagate and combine node representations across graphs. This mechanism is more complex compared with other methods that only rely on node-pair similarity measures (e.g., dot product, Euclidean distance, or cosine similarity). We believe that presenting it in equations helps readers better understand this unique approach.
>
>   Moreover, GMN, as a pioneering method in graph similarity, has influenced subsequent techniques. For instance, **H2MN** [2] adopts a similar mechanism to fuse cross-graph hyperedge representations, while **AEDNet** [3] adapts this mechanism for subgraph isomorphism prediction, proposing a unidirectional cross-propagation mechanism, in contrast to the bidirectional fusion used in the former two methods.
>
>   To further clarify the relationships between these methods, we have revised **Section 3.5 "Fine-grained Level Scoring"**. Additionally, in line with the reviewer’s suggestion, we have added **a summary table at the end of each step** to summarizes the distinct components used in the step.
>
> - **Requested Change 2:** Since the survey focuses only on three core methodologies, it can be more 'educational' i.e. by adding figures that juxtapose them with toy examples.
> - **Response:** Thank you for your suggestion. We have added **Figure 1 "Toy example"** to illustrate how three core methodologies works.
>
> - **Requested Change 3:** Similarly, in section 2.2.2, it could be meaningful to show the formulation of a bare-minimum model and how it changes to learn each of the metrics.
> - **Response:** We appreciate the reviewer’s suggestion. We believe that **Section 2.2.2 "Problem Formulation" "General Framework"** already provides the formulation for a basic model. However, the key to how the model adapts to learn each metric lies in designing models that align with the properties of the target concepts. To clarify this, we have added a discussion on the properties of each selected concept in the revised **Section 2.1 "Graph Similarity Concepts in Graph Theory."** Additionally, **Section 2.4 "GTDGSL vs. Deep Graph Similarity Learning (DGSL)"** has been added to provide a more detailed discussion on the design of the GTDGSL model.
>
> - **Requested Change 4:** Moreover, although the technical details are addressed adequately overall in sections 3.3-3.8, it is hard to follow which method pairs with which similarity technique (which to my understanding is important to get an overall sense of how the methods compare, based on the paper's introduction). A specific label or color could be helpful for this.
> - **Response:** We understand the reviewer’s concern. However, given that the paper discusses over twenty methods, using labels or colors to distinguish them may not provide clarity. Instead, in response to **Requested Change 1**, we have added a summary table at the end of each step that summarizes the key components employed. We believe these tables will address the reviewer’s concern by providing a clear and organized comparison of the methods and their associated techniques.
>
> - **Requested Change 5:** Language is redundant in some cases e.g. section 4 first and second paragraphs overlap.
> - **Response:** Thank you for pointing this out. We have carefully revised and streamlined the content to avoid unnecessary overlap.

---

> ### Author Response · Authors · 2025-02-12
> **Response Part 2**
>
> ------
>
> We sincerely appreciate your detailed and insightful feedback, which has helped us refine the paper. We believe that these changes have significantly improved the clarity and educational value of the paper. We look forward to your further feedback and hope the revisions meet your expectations.
>
> [1] Yujia Li, Chenjie Gu, Thomas Dullien, Oriol Vinyals, and Pushmeet Kohli. Graph matching networks for learning the similarity of graph structured objects. ArXiv, abs/1904.12787, 2019.
>
> [2] Zhen Zhang, Jiajun Bu, Martin Ester, Zhao Li, Chengwei Yao, Zhi Yu, and Can Wang. H2mn: Graph similarity learning with hierarchical hypergraph matching networks. In Proceedings of the 27th ACM SIGKDD Conference on Knowledge Discovery & Data Mining, pp. 2274–2284, 2021.
>
> [3] Zixun Lan, Ye Ma, Limin Yu, Linglong Yuan, and Fei Ma. Aednet: Adaptive edge-deleting network for subgraph matching. Pattern Recognition, 133:109033, 2023. ISSN 0031-3203. doi: https://doi.org/10.1016/j.patcog.2022.109033. URL https://www.sciencedirect.com/science/article/pii/
> S0031320322005131.

---

> ### Author Response · Authors · 2025-02-18
>
> Dear Reviewer dMz7,
>
> Once again, we sincerely appreciate your thoughtful and constructive feedback. As the discussion period comes to a close, we would like to confirm whether our responses align with your comments and if the revisions meet your expectations. Please let us know if you have any further suggestions. We would be happy to continue the discussion to address any remaining concerns.

---

### Review · Reviewer_uDxE · 2025-02-06

**Summary Of Contributions:**

This paper conducts a survey for graph theory-based deep graph similarity learning (GTDGSL) methods. It introduces 3 theory-based deep graph similarity measures, i.e., subgraph isomorphism (SI), maximum common subgraph (MCS), and graph edit distance (GED). The key steps of GTDGSL methods are also discussed, i.e., input preparation, preprocessing, node encoding, fine- and coarse-grained scoring, graph feature generation, and training objectives.

**Audience:**

Yes

**Claims And Evidence:**

No

**Requested Changes:**

R1: Clarify the differences between GTDGSL methods and other graph similarity learning methods. Beside the supervision signals and prediction targets, are their model designs fundamentally different? If yes, how do these designs correspond to graph theory? What are the theoretical guarantees and insights provided by the GTDGSL methods? I presume that if these methods claimed to be graph theory-based, they should have some theory properties.

R2: Provide insights, in fact, lacking insights is my main complain for the paper, and readers should have concrete takeaways from a survey rather than a set of isolated papers. To provide insights, some important questions need to be answered, what are the best methods to predict SI, MCS, GED, respectively; how good are they in terms of accuracy, training time, and inference cost? Each step of GTDGSL has many solutions, what are the advantages and disadvantages of the solutions, are there clear winners? These points should be made explicit with tables like Table 1.

R3: The presentation can be more succinct in many places. For instance, Section 2.1 introduces Graph Similarity Concepts in Graph Theory, which are text-book contents and can be made short; Section 7.1.2 repeats many contents of Section 4.

**Strengths And Weaknesses:**

Strengths

S1: Graph theory-based deep graph similarity learning (GTDGSL) methods have many important applications and emerging methods, and thus a survey for GTDGSL methods has significance.

S2: GTDGSL methods are decomposed into key steps, and the designs for each step are discussed.

S3: Overall, the paper is well-written and easy to follow.


Weakness

W1: The differences between GTDGSL methods and other graph similarity learning methods are not clear.

W2: The survey lacks insights, e.g., which methods are the SOTA, what techniques are effective for the considered problems, and how well can current graph similarity learning methods perform.

W3: The paper can be made more succinct in many places.

Please see the details in the Required Changes

---

> ### Author Response · Authors · 2025-02-12
> **Response Part 1**
>
> Dear Reviewer  uDxE,
>
> Thank you for your insightful suggestions on our paper. We appreciate your expertise and your contributions. Below is a detailed response to each of the points you raised, along with the **revisions**, which are **highlighted in violet within the revised manuscript**, we have made based on your comments:
>
> - **R1:** Clarify the differences between GTDGSL methods and other graph similarity learning methods. Beside the supervision signals and prediction targets, are their model designs fundamentally different? If yes, how do these designs correspond to graph theory? What are the theoretical guarantees and insights provided by the GTDGSL methods? I presume that if these methods claimed to be graph theory-based, they should have some theory properties.
>
> - **Response:** We thank the reviewer for the insightful suggestion. Indeed, their model designs are fundamentally different. While general graph similarity learning methods are free to explore similarity measures in a data-driven manner, GTDGSL methods are specifically designed to reflect the properties of the targeted graph similarity concept in order to ensure consistent performance. Therefore, GTDGSL designs are constrained by these properties.
>
>   For example, for GED computation under a uniform cost setting, Euclidean distance is an ideal distance function, as it satisfies properties like symmetry and the triangle inequality. However, for the subgraph isomorphism (SI), Euclidean distance may not be suitable because SI requires transitivity. Specifically, if $Q$ is a subgraph of $D^\prime$ and $D^\prime$ is a subgraph of $D$, then $Q$ must also be a subgraph of $D$. Euclidean distance does not guarantee this transitive property, as as proximity between $Q$ and $D^\prime$,  as well as between $D^\prime$ and $D$,  does not imply proximity between $Q$ and $D$.
>
>   Due to these constraints, GTDGSL methods can, in turn, provide theoretical guarantees and insights that general methods may not. For instance, the partial order nature of SI can suggest a hierarchical embedding space, a solution space that general methods cannot reliably provide. These theoretical insights stem from the alignment between the model design and the graph theory concept it targets.
>
>   In summary, in response to the reviewer's suggestion, we have revised **Section 2.1.1 "Graph Similarity Concepts in Graph Theory"** to better emphasize the relationships between the three graph similarity concepts and their distinct properties. Additionally, we have added **Section 2.4 "GTDGSL vs. Deep Graph Similarity Learning (DGSL)"** to provide a more detailed discussion on the design differences between these two approaches. We hope these revisions address the reviewer’s concerns.

---

> > ### Comment · Reviewer_uDxE · 2025-02-28
> > **Comments on response part 1**
> >
> > Thank you for the response. However, the response does NOT address my concern. The added Section 2.4 (also the response) discusses the Design Requirements for GTDGSL to have the properties of theory-based graph similarity measures. This is nice but my main concern is that how do these requirements lead to the Designs of the Models (e.g., model architecture, training sample generation method) and which Designs are Effective for which Requirements. I think the survey should make this part clear.

---

> > > ### Author Response · Authors · 2025-03-03
> > > **Further response part 1**
> > >
> > > We appreciate your patience and clarification of your concern. To address it, we have revised Section 2.4 and highlighted the changes in blue, incorporating a detailed discussion and a summary table. Below, we provide a detailed explanation.
> > >
> > > GTDGSL focuses on pairwise graph similarity, typically based on graph structure, node/edge features, and graph theory constraints, making it sensitive to small differences between graphs. In contrast, general graph similarity learning (DGSL) focuses on "semantic" or category-level similarity, where the goal is to determine whether graphs belong to the same category, requires identifying features that distinguish graphs from different categories.
> > >
> > > These differing goals lead to distinct model designs. Next, we explain their differences from two perspectives: **Model Architecture** and **Training Sample Generation Method**.
> > >
> > > **Model Architecture:**
> > >
> > > * Model Input: GTDGSL focuses on computing the relative similarity between graph pairs, which can be viewed as finding a optimal (partial) mapping between graphs. In contrast, DGSL focuses on the properties of individual graphs. This distinction leads to differences in model input formats. GTDGSL typically takes two graphs as input, whereas DGSL processes graphs individually. To enhance effectiveness, some GTDGSL methods augment the initial node or graph features with pair-dependent heuristics to provide task-specific node compatibility information.
> > > * Graph Interactions: GTDGSL often requires cross-graph interactions to capture similarity between graph pairs. It employs node/graph comparisons or cross-graph node/graph fusions to incorporate contextual information from another graph, aiding in the alignment of structural and feature information. The choice of interaction method involves trade-offs in terms of efficiency, effectiveness, and indexability. In contrast, DGSL processes a single graph without direct interaction with others.
> > > * Model Output: The output of GTDGSL is a pair-dependent similarity score. In contrast, DGSL produces a likelihood for each possible label for a given graph.
> > >
> > > **Training Sample Generation Method:**
> > >
> > > *  Training Data: The training data for GTDGSL can be sourced from graphs or sampled graphs across various domains, as GTDGSL is intended to be domain-agnostic. Graph samples for GTDGSL are typically generated through methods like BFS, DFS, or random walk-based traversals. For training graph pairs, similarity is often computed using conventional algorithms. In contrast, DGSL selects datasets based on the specific task, requiring only coarse-grained node/graph category information. This category information is typically labeled based on human-defined rules and post-hoc facts, which may change depending on the dataset or task.
> > > *  Data Augmentation: Since exact similarity computation in GTDGSL is often challenging, ground-truth supervision may be unavailable in some cases. In such instances, graph pairs or triplets (especially for subgraph isomorphism prediction) can be effectively generated by applying perturbations (e.g., node/edge additions, deletions, relabeling) to graphs. In contrast, for DGSL, since similarity measures in such tasks are typically unknown, generating reliable training data remains challenging. When data is scarce, techniques such as Masked Autoencoders and adversarial learning can be employed to generate additional samples. These methods simulate category distributions to generate in-distribution data, enhance data diversity.

---

> ### Author Response · Authors · 2025-02-12
> **Response Part 2**
>
> - **R2:** Provide insights, in fact, lacking insights is my main complain for the paper, and readers should have concrete takeaways from a survey rather than a set of isolated papers. To provide insights, some important questions need to be answered, what are the best methods to predict SI, MCS, GED, respectively; how good are they in terms of accuracy, training time, and inference cost? Each step of GTDGSL has many solutions, what are the advantages and disadvantages of the solutions, are there clear winners? These points should be made explicit with tables like Table 1.
>
> - **Response:**  We understand the reviewer's concern, but as a survey paper rather than a benchmark study, we are cautious about defining "best methods" for each task, as these models have not been systematically evaluated on the same datasets or under consistent experimental settings. However, to provide insights, we summarize findings based on reported results in the literature.
>
>   - **Accuracy.** The most recent methods targeting each concept generally show superior performance in their original paper. For instance, D2Match [1] for SI prediction, MCSNet [2] for MCS prediction, as well as GEDGNN [3], MATA* [4], and GED-CDA [5] for GED prediction have demonstrated state-of-the-art results. A common characteristic among these methods is their use of Fine-grained Level Scoring, which considers node-level similarity. However, Fine-grained Level Scoring methods are not necessarily clear winners. Experimental results also suggest that Coarse-grained Level Scoring methods often achieve competitive performance, outperforming many other Fine-grained Level Scoring methods. For example, NeuroMatch for SI prediction and Greed and Eric for GED prediction have shown strong results on some datasets.
>   - **Training and inference time.** Fine-grained Level Scoring methods typically have at least quadratic complexity. Compared with Coarse-grained Level Scoring methods, which only compare graph-level representations, Fine-grained Level Scoring ones incur significantly higher training time and inference cost as graph size increases. Additionally, some methods, such as MCSNet and GEDGNN, employ Gumbel Sinkhorn Networks, which require iterative optimization of the similarity matrix, further increasing computational complexity. Due to the irregular structure of graphs, these methods often require padding graphs to the same size for computation, and in some cases, padding all graphs in a dataset to a uniform size to enable batch training. This padding requirement further exacerbates computational overhead. Conversely, NeuroMatch and Greed, due to their simpler architectures, achieve the best training and inference efficiency. Meanwhile, Eric introduces an unsupervised graph-node alignment constraint during training, which increases training time. However, this constraint can be removed during inference, making it one of the fastest models at test time.
>
>  Given the distinct properties of the three graph similarity concepts, we do not identify a clear winner approach at each step for all cases. For example, Gumbel Sinkhorn Networks have demonstrated strong performance in GED and MCS computation, as seen in GEDGNN and MCSNet. However, their effectiveness in SI prediction is less consistent. While IsoNet achieves state-of-the-art SI prediction on small graphs (with only tens of nodes) using this optimization technique, D2Match experiments show that it struggles to converge on some datasets. We hypothesize that this discrepancy arises because Gumbel Sinkhorn Networks compute an optimal transport cost and enforce a strict node-to-node mapping, whereas SI does not require such an explicit alignment for unmatchable graphs. To make these insights more explicit, we have **added a summary at the end of each step**, providing a clear and organized comparison of the methods and their associated techniques within the step. We hope these revisions address the reviewer’s concerns and improve the clarity and takeaways of our survey.
>
> - **R3:** The presentation can be more succinct in many places. For instance, Section 2.1 introduces Graph Similarity Concepts in Graph Theory, which are text-book contents and can be made short; Section 7.1.2 repeats many contents of Section 4.
> - **Response:** We have revised **Section 2.1.1 "Graph Similarity Concepts in Graph Theory"** to minimize the definitions and reduce the descriptions of conventional algorithms, ensuring the section is more succinct.  Additionally, we have carefully streamlined the overall presentation of the paper, including **Section 7.1.2** and **Section 4**, to improve clarity and avoid redundancy.

---

> > ### Comment · Reviewer_uDxE · 2025-02-28
> > **Comments on Response Part 2**
> >
> > Thank you for the detailed response. A summary table at the end of each step indeed significantly enhances the insights.

---

> > > ### Author Response · Authors · 2025-03-03
> > > **Further response part 2**
> > >
> > > ### Summary Table:
> > >
> > > |                        | **GTDGSL**                       | **DGSL**                             | **Effective Design Choices**                                 |
> > > | ---------------------- | -------------------------------- | ------------------------------------ | ------------------------------------------------------------ |
> > > | **Focus**              | Pairwise graph similarity        | Category-level similarity            | GTDGSL requires sensitivity to small structural and feature differences, while DGSL focuses on identifying features that distinguish graphs from different categories. |
> > > | **Model Input**        | Two graphs                       | Single graph                         | GTDGSL incorporates pair-dependent features to enhance initial node/graph representations, while GSL processes graphs independently. |
> > > | **Graph Interactions** | Cross-graph interactions         | No direct interaction between graphs | GTDGSL applies node/graph comparisons or cross-graph fusion, balancing efficiency, effectiveness, and indexability. DGSL focuses on extracting per-graph representations. |
> > > | **Model Output**       | Pair-dependent similarity score  | Likelihood for each possible label   | GTDGSL enables fine-grained similarity scoring for ranking/matching, whereas DGSL provides classification probabilities. |
> > > | **Training Data**      | Graph pairs from diverse domains | Task-specific datasets               | GTDGSL computes ground truth using conventional algorithms, while DGSL datasets are labeled based on human-defined rules or post-hoc facts. |
> > > | **Data Augmentation**  | Perturbation-based sampling      | In-distribution  data generation     | GTDGSL requires carefully curated synthetic data to reflect specific similarity properties, while DGSL relies on data augmentation or adversarial methods to simulate category distributions. |
> > >
> > > We hope this revision correctly aligns with your suggestion. Again, we sincerely thank you for your engagement and valuable contributions to improving the quality of our submission.

---

> ### Author Response · Authors · 2025-02-12
> **Response Part 3**
>
> ------
>
> Again, we sincerely appreciate your insightful feedback, which definitely has helped us refine the paper. We look forward to your further feedback and hope the revisions meet your expectations.
>
> [1] Xuanzhou Liu, Lin Zhang, Jiaqi Sun, Yujiu Yang, and Haiqin Yang. D2Match: Leveraging deep learning and degeneracy for subgraph matching. In Andreas Krause, Emma Brunskill, Kyunghyun Cho, Barbara Engelhardt, Sivan Sabato, and Jonathan Scarlett (eds.), Proceedings of the 40th International Conference on Machine Learning, volume 202 of Proceedings of Machine Learning Research, pp. 22454–22472. PMLR, 23–29 Jul 2023b. URL https://proceedings.mlr.press/v202/liu23ba.html.
>
> [2] Indradyumna Roy, Soumen Chakrabarti, and Abir De. Maximum common subgraph guided graph retrieval: Late and early interaction networks. ArXiv, abs/2210.11020, 2022a.
>
> [3] Chengzhi Piao, Tingyang Xu, Xiangguo Sun, Yu Rong, Kangfei Zhao, and Hong Cheng. Computing graph edit distance via neural graph matching. Proc. VLDB Endow., 16(8):1817–1829, 2023. doi: 10.14778/3594512.3594514. URL https://www.vldb.org/pvldb/vol16/p1817-cheng.pdf.
>
> [4] Junfeng Liu, Min Zhou, Shuai Ma, and Lujia Pan. Mata*: Combining learnable node matching with a\*algorithm for approximate graph edit distance computation. In Proceedings of the 32nd ACM International Conference on Information and Knowledge Management, CIKM ’23, pp. 1503–1512, New York, NY, USA, 2023a. Association for Computing Machinery. ISBN 9798400701245. doi: 10.1145/3583780.3614959. URL https://doi.org/10.1145/3583780.3614959.
>
> [5] Ruiqi Jia, Xianbing Feng, Xiaoqing Lyu, and Zhi Tang. Graph-graph context dependency attention for graph edit distance. In ICASSP 2023 - 2023 IEEE International Conference on Acoustics, Speech and Signal Processing (ICASSP), pp. 1–5, 2023. doi: 10.1109/ICASSP49357.2023.10094975.
>
> ------

---

> ### Author Response · Authors · 2025-02-18
>
> Dear Reviewer uDxE,
>
> Once again, we sincerely appreciate your insightful suggestions. As the discussion period draws to a close, we would like to confirm whether our responses align with your comments and if the revisions meet your expectations. Please let us know if you have any further suggestions. We would be happy to continue the discussion to address any remaining concerns.

---

### Decision · Action_Editor_xjVM · 2025-03-19

**Recommendation:** Accept with minor revision

**Comment:**

This paper provides a comprehensive survey on graph theory-based deep graph similarity learning (GTDGSL) methods, including discussions on subgraph isomorphism (SI), maximum common subgraph (MCS), and graph edit distance (GED). It delineates the key steps in GTDGSL pipelines, such as input preparation, preprocessing, node encoding, scoring methods, graph feature generation, and training objectives. The paper is well-written, well-organized, and provides an overview of the developments in GTDGSL methods along with their applications and challenges.

However, improvements are needed to address some clarity issues, enhance the insights provided, and incorporate more critical analysis based on the feedback from reviewers.

1. **Clarify Key Differences and Effective Design Choices**
   A primary concern raised by Reviewer uDxE is the need for greater clarity on how GTDGSL methods differ fundamentally from other graph similarity learning methods. This includes not only their constraints rooted in graph theory but also how those constraints influence model architecture and training sample generation. To address this, Section 2.4 should be revised to include a detailed breakdown of these design connections, with a summary table highlighting effective design choices for specific theoretical requirements.

2. **Enhance Insights and Summary Comparisons**
   Both Reviewers uDxE and dMz7 emphasized the need for actionable insights rather than isolated descriptions of methods. To address this, summary tables should be added at the end of each step in Section 3 to compare methods in terms of accuracy, training time, computational efficiency, and use cases. These tables will make it easier to identify trade-offs between approaches, such as Fine-grained vs. Coarse-grained scoring methods, and provide readers with concrete takeaways.

3. **Include Critical Analyses and Practical Relevance**
   Reviewer ofZ6 highlighted the need for a critical analysis of model limitations (e.g., GNNs’ over-smoothing and scalability issues) and real-world case studies to demonstrate practical relevance. Section 6 should be expanded with concise case studies for each application domain and discuss how SI, MCS, and GED can be combined in broader pipelines. Additionally, Section 2.2.1 should analyze GNN limitations and how GTDGSL methods address these challenges.

By implementing these revisions, the paper will become more insightful, concise, and practically relevant, addressing the core concerns of all reviewers.

**Audience:**

This paper is expected to provide valuable insights for the graph neural network domain, a crucial and rapidly evolving area of deep learning.

**Claims And Evidence:**

This paper systematically compares existing graph theory-based deep similarity learning (GTDGSL) methods, highlighting their commonalities and differences in training pipelines and techniques. Additionally, it discusses the key challenges, applications, and future research directions in this domain.

---

> ### Author Response · Authors · 2025-05-01
>
> Dear Action Editor and Reviewers ,
>
> We are deeply grateful for the time, expertise, and thoughtful guidance you have provided throughout the review process. Your constructive feedback has been instrumental in strengthening our work, and we sincerely appreciate your commitment to advancing the quality of scholarly research.
>
> Thank you again for your invaluable contributions.
>
> Best regards,
>
> Authors